# MISSPECIFIED $Q$-LEARNING WITH SPARSE LINEAR FUNCTION APPROXIMATION: TIGHT BOUNDS ON APPROXIMATION ERROR

**Ally Yalei Du**
Carnegie Mellon University
aydu@andrew.cmu.edu

**Lin F. Yang**
University of California, Los Angeles
linyang@ee.ucla.edu

**Ruosong Wang**
CFCS and School of Computer Science
Peking University
ruosongwang@pku.edu.cn

## ABSTRACT

The recent work by Dong & Yang (2023) showed for misspecified sparse linear bandits, one can obtain an $O(\epsilon)$-optimal policy using a polynomial number of samples when the sparsity is a constant, where $\epsilon$ is the misspecification error. This result is in sharp contrast to misspecified linear bandits without sparsity, which require an exponential number of samples to get the same guarantee. In order to study whether the analog result is possible in the reinforcement learning setting, we consider the following problem: assuming the optimal $Q$-function is a $d$-dimensional linear function with sparsity $s$ and misspecification error $\epsilon$, whether we can obtain an $O(\epsilon)$-optimal policy using number of samples polynomially in the feature dimension $d$. We first demonstrate why the standard approach based on Bellman backup or the existing optimistic hypothesis class elimination approach such as OLIVE Jiang et al. (2017) achieves suboptimal guarantees for this problem. We then design a novel elimination-based algorithm to show one can obtain an $O(H\epsilon)$-optimal policy with sample complexity polynomially in the feature dimension $d$ and planning horizon $H$. Lastly, we complement our upper bound with an $\widetilde{\Omega}(H\epsilon)$ suboptimality lower bound, giving a complete picture of this problem.

## 1 INTRODUCTION

Bandit and reinforcement learning (RL) problems in real-world applications, such as autonomous driving (Kiran et al., 2021), healthcare (Esteva et al., 2019), recommendation systems (Bouneffouf et al., 2012), and advertising (Schwartz et al., 2017), face challenges due to the vast state-action space. To tackle this, function approximation frameworks, such as using linear functions or neural networks, have been introduced to approximate the value functions or policies. However, real-world complexities often mean that function approximation is agnostic; the function class captures only an approximate version of the optimal value function, and the misspecification error remains unknown. A fundamental problem is understanding the impact of agnostic misspecification errors in RL.

Prior works show even minor misspecifications can lead to exponential (in dimension) sample complexity in the linear bandit settings (Du et al., 2020; Lattimore et al., 2020) if the goal is to learn a policy within the misspecification error. That is, finding an $O(\epsilon)$-optimal action necessitates at least $\Omega(\exp(d))$ queries (or samples) to the environment, where $\epsilon$ is the misspecification error. Recently, Dong & Yang (2023) demonstrated that by leveraging the sparsity structure of ground-truth parameters, one can overcome the exponential sample barrier in the linear bandit setting. They showed that with sparsity $k$ in the ground-truth parameters, it is possible to learn an $O(\epsilon)$-optimal action with only $O\left((d/\epsilon)^k\right)$ samples. In particular, when $k$ is a constant, their algorithm achieves a polynomial sample complexity guarantee.

A natural question is whether we can obtain a similar sample complexity guarantee in the RL setting. This question motivates us to consider a more general question:

*Given a that the $Q^*$ function is a $d$-dimensional linear function with sparsity $k$ and misspecification error $\epsilon$, can we learn an $O(\epsilon)$-optimal policy using $\mathrm{poly}(d, 1/\epsilon)$ samples, when $k$ is a constant?[1]*

It turns out that by studying this question, we obtain a series of surprising results which cannot be explained by existing RL theories.

## 1.1 OUR CONTRIBUTIONS

In this paper, we propose an RL algorithm that can handle linear function approximation with sparsity structures and misspecification errors. We also show that the suboptimality achieved by our algorithm is near-optimal, by proving information-theoretic hardness results. Here we give a more detailed description of our technical contributions.

**Our Assumption.** Throughout this paper, we assume the RL algorithm has access to a feature map where, for each state-action pair $(s, a)$, we have the feature $\phi(s, a)$ with $\|\phi(s, a)\| \leq 1$. We make the following assumption, which states that there exists a sequence of parameters $\theta^* = (\theta_0^*, \ldots, \theta_{H-1}^*)$ where each $\theta_h^* \in \mathbb{S}^{d-1}$ is $k$-sparse, that approximates the optimal $Q$-function up to an error of $\epsilon$.

**Assumption 1.** *There exists $\theta^* = (\theta_0^*, \ldots, \theta_{H-1}^*)$ where each $\theta_h^* \in \mathbb{S}^{d-1}$ is $k$-sparse, such that*

$$|\langle \phi(s, a), \theta_h^* \rangle - Q^*(s, a)| \leq \epsilon$$

*for all $h \in [H]$, all states $s$ in level $h$, and all actions $a$ in the action space.*

When $H = 1$, Assumption 1 is equivalent to the bandit setting in Dong & Yang (2023). Note that we have a different optimal parameter $\theta_h$ for each level $h \in [H]$.

We can approximate $\theta^*$ using an $\epsilon$-net of the sphere $\mathbb{S}^{k-1}$ and the set of all $k$-sized subset of $[d]$. Therefore, when $k$ is a constant, we may assume that each $\theta_h^*$ lies in a set with size polynomial in $d$. Then, a natural idea is to enumerate all possible policies induced by the parameters in that finite set, and choose the one with the highest cumulative reward. However, although the number of parameter candidates in each individual level has polynomial size, the total number of induced policies would be exponential in $H$, and the sample complexity of such approach would also be exponential in $H$.

**The Level-by-level Approach.** Note that when the horizon length $H = 1$, the problem under consideration is equivalent to a bandit problem, which can be solved by previous approaches (Dong & Yang, 2023). For the RL setting, a natural idea is to first apply the bandit algorithm in Dong & Yang (2023) on the last level, and then apply the same bandit algorithm on the second last level based on previous results and Bellman-backups, and so on. However, we note that to employ such an approach, the bandit algorithm needs to provide a "for-all" guarantee, i.e., finding a parameter that approximates the rewards of all arms, instead of just finding a near-optimal arm. On the other hand, existing bandit algorithms will amplify the approximation error of the input parameters by a constant factor, in order to provide a for-all guarantee. Concretely, existing bandit algorithms can only find a parameter $\theta$ so that $\theta$ approximates the rewards of all arms by an error of $2\epsilon$. As we have $H$ levels in the RL setting, the final error would be exponential in $H$, and therefore, such a level-by-level approach would result in a suboptimality that is exponential in $H$.

One may ask if we can further improve existing bandit algorithms, so that we can find a parameter $\theta$ that approximates the rewards of all arms by an error of $\epsilon$ plus a statistical error that can be made arbitrarily small, instead of $2\epsilon$. The following theorem shows that this is information-theoretically impossible unless one pays a sample complexity proportional to the size of the action space.

**Theorem 1.1.** *Under Assumption 1 with $d = k = 1$, any bandit algorithm that returns an estimate $\hat{r}$ such that $|\hat{r}(a) - r(a)| < 2\epsilon$ for all arms $a$ with probability at least $0.95$ requires at least $0.9n$ samples, where $n$ is the total number of arms.*

Therefore, amplifying the approximation error by a factor of $2$ is not an artifact of existing bandit algorithms. Instead, it is information-theoretically impossible.

---

[1]Strictly speaking, as will be demonstrated in this paper, the best one can do is an $O(\epsilon H)$-optimal policy where $H$ is the horizon length.

Geometric error amplification is a common issue in the design of RL algorithm with linear function approximation (Zanette et al., 2019; Weisz et al., 2021; Wang et al., 2020a; 2021a). It is interesting (and also surprising) that such an issue arises even when the function class has sparsity structures.

**Optimistic Value Function Elimination.** Another approach for the design of RL algorithm is based on optimistic value function elimination. Such an approach was proposed by Jiang et al. (2017) and was then generalized to broader settings (Sun et al., 2019; Du et al., 2021; Jin et al., 2021; Chen et al., 2022b). At each iteration of the algorithm, we pick the value functions in the hypothesis class with maximized value. We then use the induced policy to collect a dataset, based on which we eliminate a bunch of value functions from the hypothesis class and proceed to the next iteration.

When applied to our setting, existing algorithms and analysis achieve a suboptimality that depends on the size of the parameter class, which could be prohibitively large. Here, we use the result in Jiang et al. (2017) as an example. The suboptimality of their algorithm is $H\sqrt{M}\epsilon$, where $M$ is *Bellman rank* of the problem. For our setting, we can show that there exists an MDP instance and a feature map that satisfies Assumption 1, whose induced Bellman rank is large.

**Proposition 1.2.** *There exists an MDP instance $\mathcal{M} = (\mathcal{S}, \mathcal{A}, H, P, r)$ with $|\mathcal{A}| = 2$, $H = \log d$, $|\mathcal{S}| = d - 1$, with $d$-dimensional feature map $\phi$ satisfying Assumption 1 with $k = 1$, such that its Bellman rank is $d$.*

Given Proposition 1.2, if one naïvely applies the algorithm in Jiang et al. (2017), the suboptimality would be $O(H\sqrt{d}\epsilon)$ in our setting, which necessitates new algorithm and analysis. In Section 4, we design a new RL algorithm whose performance is summarized in the following theorem.

**Theorem 1.3.** *Under Assumption 1, with probability at least $1 - \delta$, Algorithm 1 returns a policy with suboptimality at most $(4\epsilon_{\text{stat}} + 2\epsilon_{\text{net}} + 2\epsilon)H$ by taking $O(kd^k H^3 \cdot \ln(dH/\epsilon_{\text{net}}\delta) \cdot \epsilon_{\text{net}}^{-k}\epsilon_{\text{stat}}^{-2})$ samples.*

Here $\epsilon_{\text{stat}}$ is the statistical error. Compared to the existing approaches, Theorem 1.3 achieves a much stronger suboptimality guarantee. Later, we will also show that such a guarantee is near-optimal.

Although based on the same idea of optimistic value function elimination, our proposed algorithm differs significantly from existing approaches (Jiang et al., 2017; Sun et al., 2019; Du et al., 2021; Jin et al., 2021; Chen et al., 2022b) to exploit the sparsity structure. While existing approaches based on optimistic value function elimination try to find a sequence of parameters that maximize the value of the initial states, our new algorithm selects a parameter that maximizes the empirical roll-in distribution at all levels. Also, existing algorithms eliminate a large set of parameters in each iteration, while we only eliminate the parameters selected during the current iteration in our algorithm.

These two modifications are crucial for obtaining a smaller suboptimality guarantee, smaller sample complexity, and shorter running time. In existing algorithms, parameters at different levels are interdependent, i.e. the choice of parameter at level $h$ affects the choice of parameter at level $h + 1$. Our new algorithm simplifies this by maintaining a parameter set for each level, so each level operates independently. Further, we can falsify and eliminate any parameter showing large Bellman error at any level $h$, since otherwise we would have found another parameter with larger induced value function at level $h + 1$ to make the error small. Consequently, since we eliminate at least one parameter at each iteration, we obtain fewer iterations and enhanced sample complexity.

**The Hardness Result.** One may wonder if the suboptimality guarantee can be further improved. In Section 3, we show that the suboptimality guarantee by Theorem 1.3 is near-optimal.

We first consider a weaker setting where the algorithm is not allowed to take samples, and the function class contains a single sequence of functions. I.e., we are given a function $\hat{Q} : \mathcal{S} \times \mathcal{A} \to \mathbb{R}$, such that $|\hat{Q}(s, a) - Q^*(s, a)| \leq \epsilon$ for all $(s, a) \in \mathcal{S} \times \mathcal{A}$.

We show that for this weaker setting, simply choosing the greedy policy with respect to $\hat{Q}$, which achieves a suboptimality guarantee of $O(H\epsilon)$, is actually optimal. To prove this, we construct a hard instance based on binary tree. Roughly speaking, the optimal action for each level is chosen uniformly random from two actions $a_1$ and $a_2$. At all levels, the reward is $\epsilon$ if the optimal action is chosen, and os 0 otherwise. For this instance, there exists a fixed $\hat{Q}$ that provides a good approximation to the optimal $Q$-function, regardless of the choice of the optimal actions. Therefore, $\hat{Q}$ reveals

no information about the optimal actions, and the suboptimality of the returned policy would be at least $\Omega(H\epsilon)$. The formal construction and analysis and construction will be given in Section 3.1.

When the algorithm is allowed to take samples, we show that in order to achieve a suboptimality guarantee of $H/C\epsilon$, any algorithm requires $\exp(\Omega(C))$ samples, even when Assumption 1 is satisfied with $d = k = 1$. Therefore, for RL algorithms with polynomial sample complexity, the suboptimality guarantee of Theorem 1.3 is tight up to log factors.

To prove the above claim, we still consider the setting where $d = k = 1$, i.e., a good approximation to the $Q$-function is given to the algorithm. We also use a more complicated binary tree instance, where we divide all the $H$ levels into $H/C$ blocks, each containing $C$ levels. For each block, only one state-action pair at the last level has a reward of $\epsilon$, and all other state-action pairs in the block has a reward of $0$. Therefore, the value of the optimal policy would be $H/C \cdot \epsilon$ since there are $H/C$ blocks in total. We further show that there is a fixed function $\hat{Q}$, which provides a good approximation to the optimal $Q$-function universally for all instances under consideration.

Since $\hat{Q}$ reveals no information about the state-action pair with $\epsilon$ reward for all blocks, for an RL algorithm to return a policy with a non-zero value, it must search for a state-action pair with non-zero reward in a brute force manner, which inevitably incurs a sample complexity of $\exp(\Omega(C))$ since each block contains $C$ levels and $\exp(\Omega(C))$ state-action pairs at the last level. The formal construction and analysis and construction will be given in Section 3.2.

## 1.2 RELATED WORK

A series of studies have delved into MDPs that can be represented by linear functions of predetermined feature mappings, achieving sample complexity or regret that depends on the feature mapping's dimension. This includes linear MDPs, studied in Jin et al. (2020); Wang et al. (2019); Neu & Pike-Burke (2020), where both transition probabilities and rewards are linear functions of feature mappings on state-action pairs. Zanette et al. (2020a;b) examines MDPs with low inherent Bellman error, indicating value functions that are almost linear with respect to these mappings. Another focus is on linear mixture MDPs (Modi et al., 2020; Jia et al., 2020; Ayoub et al., 2020; Zhou et al., 2021; Cai et al., 2020), characterized by transition probabilities that combine several basis kernels linearly. While these studies often assume known feature vectors, Agarwal et al. (2020) investigates a more challenging scenario where both features and parameters of the linear model are unknown.

The literature has also witnessed a substantial surge of research in understanding how function general approximations can be applied efficiently in the reinforcement learning setting (Osband & Van Roy, 2014; Sun et al., 2019; Ayoub et al., 2020; Wang et al., 2020b; Foster et al., 2021; Chen et al., 2022b;a; Zhong et al., 2022; Foster et al., 2023; Wagenmaker & Foster, 2023; Zhou & Gu, 2022; Jiang et al., 2017; Wang et al., 2020b; Du et al., 2021; Jin et al., 2021; Kong et al., 2021; Dann et al., 2021; Zhong et al., 2022; Liu et al., 2023; Agarwal et al., 2023). To obtain good sample, error, or regret bounds, these approaches typically impose benign structures on values, models, or policies, along with benign misspecification. Amongst these works, Jiang et al. (2017) is particularly related to our work as their elimination-based algorithm, OLIVE, can be directly applied to our setting. However, as mentioned in Section 1.1, the suboptimality guarantee of their algorithm is significantly worse than our result.

In another line of works, Du et al. (2020); Dong & Yang (2023); Lattimore et al. (2020) specifically focuses on understanding misspecification in bandit and RL scenarios. Du et al. (2020) illustrated that to find an $O(\epsilon)$-optimal policy in reinforcement learning with $\epsilon$-misspecified linear features, an agent must sample an exponential (in $d$) number of trajectories, applicable to both value-based and model-based learning. Relaxing this goal, Lattimore et al. (2020) indicated that $\text{poly}(d/\epsilon)$ samples could suffice to secure an $O(\epsilon\sqrt{d})$-optimal policy in a simulator model setting of RL, though achieving a policy with an error better than $O(\epsilon\sqrt{d})$ would still require an exponential sample size. Recently, Dong & Yang (2023) introduced a solution, showing that incorporating structural information like sparsity in the bandit instance could address this issue, making it feasible to attain $O(\epsilon)$ with $O((d/\epsilon)^k)$ sample complexity, which is acceptable when the sparsity $k$ is small. Another recent independent work (Amortila et al., 2024) also obtains a suboptimality guarantee of $O(H\epsilon)$. However, their result depends on a coverability assumption and uses a different technique called disagreement-based regression (DBR), which is distinct from our assumption and techniques.

## 2 PRELIMINARIES

Throughout the paper, for a given positive integer $n$, we use $[n]$ to denote the set $\{0, 1, 2, \ldots, n-1\}$. In addition, $f(n) = O(g(n))$ denotes that there exists a constant $c > 0$ such that $|f(n)| \leq c|g(n)|$. $f(n) = \Omega(g(n))$ denotes that there exists a constant $c > 0$ such that $|f(n)| \geq c|g(n)|$. For a set $S$, $\Delta(S)$ represents the set of all probability distributions defined over $S$.

### 2.1 REINFORCEMENT LEARNING

Let $\mathcal{M} = \{\mathcal{S}, \mathcal{A}, H, P, r\}$ be a Markov Decision Process (MDP) where $\mathcal{S}$ is the state space, $\mathcal{A}$ is the action space, $H \in \mathbb{Z}_+$ is the planning horizon, $P : \mathcal{S} \times \mathcal{A} \to \Delta(\mathcal{S})$ is the transition kernel which takes a state-action pair as input and returns a distribution over states, $r : \mathcal{S} \times \mathcal{A} \to \Delta([0, 1])$ is the reward distribution. We assume $\sum_{h \in [H]} r_h \in [0, 1]$ almost surely. For simplicity, throughout this paper, we assume the initial state $s_0$ is deterministic. To streamline our analysis, for each $h \in [H]$, we use $\mathcal{S}_h \subseteq \mathcal{S}$ to denote the set of states at level $h$, and assume $\mathcal{S}_h$ do not intersect with each other.

A policy $\pi : \mathcal{S} \to \mathcal{A}$ chooses an action for each state, and may induce a trajectory denoted by $(s_0, a_0, r_0, \ldots, s_{H-1}, a_{H-1}, r_{H-1})$, where $s_{h+1} \sim P(s_h, a_h)$, $a_h = \pi(s_h)$, and $r_h \sim r(s_h, a_h)$ for all $h \in [H]$. Given a policy $\pi$ and $h \in [H]$, for a state-action pair $(s, a) \in \mathcal{S}_h \times \mathcal{A}$, the $Q$-function and value function is defined as

$$Q^\pi(s, a) = \mathbb{E}\left[\sum_{h'=h}^{H-1} r(s_{h'}, a_{h'})|s_h = s, a_h = a, \pi\right], V^\pi(s) = \mathbb{E}\left[\sum_{h'=h}^{H-1} r(s_{h'}, a_{h'})|s_h = s, \pi\right].$$

We use $V^\pi$ to denote the value of the policy $\pi$, i.e., $V^\pi = V^\pi(s_0)$. We use $\pi^*$ to denote the optimal policy. For simplicity, for a state $s \in \mathcal{S}$, we define $V^*(s) = V^{\pi^*}(s)$, and for a state-action pair $(s, a) \in \mathcal{S} \times \mathcal{A}$, we define $Q^*(s, a) = Q^{\pi^*}(s, a)$. The suboptimality of a policy $\pi$ is defined as the difference between the value of $\pi$ and that of $\pi^*$, i.e. $V^* - V^\pi$.

For any sequence of $k$-sparse parameter $\theta = (\theta_0, \ldots, \theta_{H-1})$, we define $\pi_\theta$ to be the greedy strategy based on $\theta$. In other words, for each $h \in [H]$, for a state $s \in \mathcal{S}_h$, $\pi_\theta(s) = \arg\max_{a \in \mathcal{A}} \langle \phi(s, a), \theta_h \rangle$. For each $h \in [H]$, a parameter $\theta_h$, and a state $s \in \mathcal{S}_h$, we also write $V_{\theta_h}(s) = \max_{a \in \mathcal{A}} \langle \phi(s, a), \theta_h \rangle$.

We will prove lower bounds for deterministic systems, i.e., MDPs with deterministic transition $P$ and deterministic reward $r$. In this setting, $P$ and $r$ can be regarded as functions rather than distributions. Since deterministic systems can be considered as a special case for general stochastic MDPs, our lower bounds still hold for general MDPs.

**Interacting with an MDP.** An RL algorithm takes the feature function $\phi$ and sparsity $k$ as the input, and interacts with the underlying MDP by taking samples in the form of a trajectory. To be more specific, at each round, the RL algorithm decides a policy $\pi$ and receives a trajectory $(s_0, a_0, r_0, \ldots, s_{H-1}, a_{H-1}, r_{H-1})$ as feedback. Here one trajectory corresponds to $H$ samples. We define the total number of samples required by an RL algorithm as its *sample complexity*. Our goal is to design an algorithm that returns a near-optimal policy while minimizing its sample complexity.

**The Bandits Setting.** In this paper, we also consider the bandit setting, which is equivalent to an MDP with $H = 1$. Let $\mathcal{A}$ be the action space, and $r : \mathcal{A} \to \Delta([0, 1])$ be the reward distribution. At round $t$, the algorithm chooses an action $a_t \in \mathcal{A}$ and receives a reward $r_t \sim r(a_t)$. In this case, Assumption 1 asserts that there exists $\theta^*$, such that $|\langle \phi(a), \theta^* \rangle - \mathbb{E}[r(a)]| \leq \epsilon$ for all $a \in \mathcal{A}$.

## 3 HARDNESS RESULTS

We prove our hardness results. In Section 3.1, we prove that the suboptimality of any RL algorithm is $\Omega(H\epsilon)$ if the algorithm is not allowed to take samples. This serves as a warmup for the more complicated construction in Section 3.2, where we show that for any $C$ satisfying $1 \leq 2C \leq H$, any RL algorithm requires $\exp(\Omega(C))$ samples in order to achieve a suboptimality of $\Omega(H/C \cdot \epsilon)$.

### 3.1 WARMUP: HARDNESS RESULT FOR RL WITHOUT SAMPLES

We prove that the suboptimality of any RL algorithm without sample is $\Omega(H\epsilon)$. Specifically, we consider a setting where the feature $\phi$ is 1-dimensional and equal to the optimal Q-value with an

error of $\epsilon$. This provides a simplified scenario where an approximate optimal Q-function is readily available to the algorithm. Theorem 3.1 suggests that even in such a simplified context, the best achievable suboptimality is $O(H\epsilon)$.

**Theorem 3.1.** *Given a MDP instance satisfying Assumption 1, the suboptimality of the policy returned by any RL algorithm is $\Omega(H\epsilon)$ with a probability of $0.99$ if the algorithm is not allowed to take samples. This holds even when the dimension and sparsity satisfies $d = k = 1$ and the underlying MDP is a deterministic system.*

The formal proof of Theorem 3.1 is given in Section A.1 of the Supplementary Material. Below we give the construction of the hard instance (illustration of the hard instance is given in Appendix A.1), together with an overview of the hardness proof.

Our hardness result is based on a binary tree instance. There are $H$ levels of states, and level $h \in [H]$ contains $2^h$ distinct states. Thus we have $2^H - 1$ states in total. We use $s_0, ..., s_{2^H-2}$ to denote all the states, where $s_0$ is the unique state at level 0, and $s_1, s_2$ are the states at level 1, etc. Equivalently, $\mathcal{S}_h = \{s_{2^h-1}, \ldots, s_{2^{h+1}-2}\}$. The action space $\mathcal{A}$ contains two actions, $a_1$ and $a_2$. For each $h \in [H-1]$, a state $s_i \in \mathcal{S}_h$, we have $P(s_i, a_1) = s_{2i+1}$ and $P(s_i, a_2) = s_{2i+2}$. For each $h \in [H]$, there exists an action $a_h^* \in \{a_1, a_2\}$, such that $\pi^*(s) = a_h^*$ for all $s \in \mathcal{S}_h$. Based on $a_0^*, a_1^*, \ldots, a_{H-1}^*$, for a state $s \in \mathcal{S}_h$, we define the reward function as $r(s, a) = \epsilon$ if $a = a_h^*$ and $r(s, a) = 0$ otherwise. The corresponding Q-function is $Q^*(s, a) = (H - h)\epsilon$ if $a = a_h^*$ and $Q^*(s, a) = (H - h - 1)\epsilon$ otherwise.

Now we define the 1-dimensional feature function $\phi$. For each $h \in [H]$, for all $(s, a) \in \mathcal{S}_h \times \mathcal{A}$, $\phi(s, a) = (H - h - 1)\epsilon$. Clearly, by taking $\theta^* = 1$, Assumption 1 is satisfied for our $\phi$. This finishes the construction of our hard instance.

Since the RL algorithm is not allowed to take samples, the only information that the algorithm receives is the feature function $\phi$. However, $\phi$ is always the same no matter how we set $a_0^*, a_1^*, \ldots, a_{H-1}^*$, which means the RL algorithm can only output a fixed policy. On the other hand, if $a_h^*$ is drawn uniformly at random from $\{a_1, a_2\}$, for any fixed policy $\pi$, its expected suboptimality will be $H\epsilon/2$, which proves Theorem 3.1. Our formal proof in Section A.1 of the Supplementary Material is based on Yao's minimax principle in order to cope with randomized algorithms.

## 3.2 Hardness Result for RL with Samples

In this section, we show that for any $1 \le 2C \le H$, any RL algorithm requires $\exp(\Omega(C))$ samples in order to achieve a suboptimality of $\Omega(H/C \cdot \epsilon)$.

**Theorem 3.2.** *Given a RL problem instance satisfying Assumption 1 with misspecification $\epsilon < 1/H$ and let $C \in \mathbb{R}$ such that $1 \le 2C \le H$. Any algorithm that returns a policy with suboptimality less than $H/(2C) \cdot \epsilon$ with probability at least $0.9$ needs least $0.1 \cdot C \cdot 2^C$ samples. This holds even when the dimension and sparsity satisfies $d = k = 1$ and the underlying MDP is a deterministic system.*

In the remaining part of this section, we give an overview of the proof of Theorem 3.2. We first define the MULTI-INDEX-QUERY problem.

**Definition 1.** *(MULTI-INDEX-QUERY) In the $m$-INDQ$_n$ problem, we have a sequence of $m$ indices $(i_0^*, i_1^*, \ldots, i_{m-1}^*) \in [n]^m$. In each round, the algorithm guesses a pair $(j, i) \in [m] \times [n]$ and queries whether $i = i_j^*$. The goal is to output $(j, i_j^*)$ for any $j \in [m]$, using as few queries as possible.*

**Definition 2.** *($\delta$-correct algorithm) For $\delta \in (0, 1)$, we say a randomized algorithm $A$ is $\delta$-correct for $m$-INDQ$_n$ if for any $i^* = \{i_j^*\}_{j \in [m]}$, with probability at least $1 - \delta$, $A$ outputs $(j, i_j^*)$ for some $j$.*

We first prove a query complexity lower bound for solving $m$-INDQ$_n$.

**Lemma 3.3.** *Any $0.1$-correct algorithm that solves $m$-INDQ$_n$ requires at least $0.9n$ queries.*

Our proof is based on Yao's minimax principle (Yao, 1977). See Section A.2 for the full proof.

Now we give the construction of our hard instance, together with the high-level intuition of our hardness proof. For simplicity, here we assume $C$ is an integer that divides $H$.

**The Hard Instance.** Again, our hardness result is based on a binary tree instance. The state space, action space, and the transition kernel of our hard instance are exactly the same as the instance in

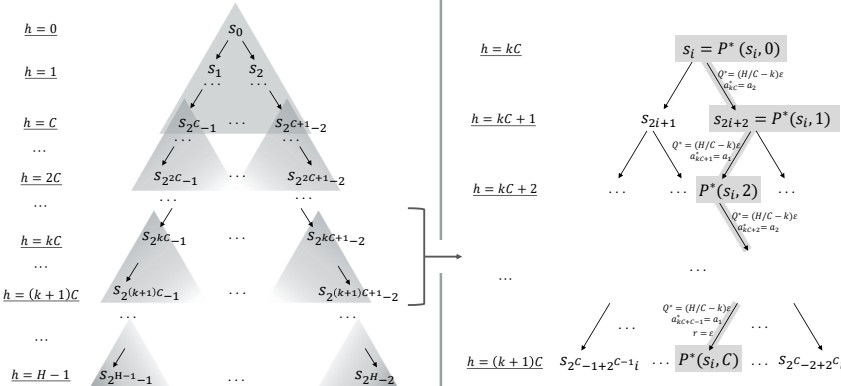

Figure 1: Illustration of the hard instance for Theorem 3.2.

Section 3.1. Moreover, similar to the instance in Section 3.1, for each $h \in [H]$, there exists an action $a_h^* \in \{a_1, a_2\}$, such that $\pi^*(s) = a_h^*$ for all $s \in \mathcal{S}_h$.

To define the reward function $r$, we first define an operator $P^*$, which can be seen as applying the transition kernel for multiple steps by following the optimal policy. For some $q \in [H/C]$, a state $s \in \mathcal{S}_{kC}$, and an integer $c \in [C]$, define $P^*(s, c) = s$ if $c = 0$ and $P^*(s, c) = P(P^*(s, c-1), a_{qC+c-1}^*)$ otherwise. The reward function $r(s, a)$ is then defined to be $\epsilon$ if $s = P^*(s', C-1)$ for some $s' \in \mathcal{S}_{qC}$ where $q \in [H/C]$ and $a = a_{qC+C-1}^*$. For all other $(s, a) \in \mathcal{S} \times \mathcal{A}$, we define $r(s, a) = 0$. Accordingly, for each $(q, c) \in [H/C] \times [C]$, $s \in \mathcal{S}_{qC+c}$, and $a \in \mathcal{A}$, we have $Q^*(s, a) = (H/C - q)\epsilon$ if $s = P^*(s', c)$ for some $s' \in \mathcal{S}_{qC}$ and $a = a_{qC+c}^*$. For all other $(s, a) \in \mathcal{S} \times \mathcal{A}$, we have $Q^*(s, a) = (H/C - q - 1)\epsilon$. This also implies that the value of the optimal policy is $H/C \cdot \epsilon$.

We define the 1-dimensional feature function $\phi$ such that, for each $(q, c) \in [H/C] \times [C]$, $s \in \mathcal{S}_{qC+t}$ and $a \in \mathcal{A}$, $\phi(s, a) = (H/C - q)\epsilon$. Clearly, Assumption 1 is satisfied when taking $\theta^* = 1$. This finishes the construction of our hard instance. An illustration is given in Figure 1.

**The Lower Bound.** Now we show that for our hard instance, if there is an RL algorithm that returns a policy with suboptimality less than $H/C \cdot \epsilon$, then there is an algorithm that solves $m$-INDQ$_n$ with $n = 2^C$ and $m = H/C$. Therefore, the correctness of Theorem 3.2 is implied by Lemma 3.3.

We first note that there exists a bijection between $\{a_1, a_2\}^C$ and $[2^C]$. We use $g : [2^C] \to \{a_1, a_2\}^T$ to denote such a bijection. Given an instance of $m$-INDQ$_n$ with $n = 2^C$ and $m = H/C$, for each $q \in [H/C]$, we set $(a_{qC}^*, a_{qC+1}^*, \ldots, a_{(q+1)C-1}^*) = g(i_q^*)$, where $(i_0^*, i_1^*, i_2^*, \ldots, i_{H/C-1}^*)$ are the target indices in the instance of $m$-INDQ$_n$. Each time the RL algorithm samples a trajectory $(s_0, a_0, r_0, \ldots, s_{H-1}, a_{H-1}, r_{H-1})$, we make $H/C$ sequential queries $(0, i_0), (1, i_1), \ldots, (H/C - 1, i_{H/C-1})$ to $m$-INDQ$_n$, where for each $q \in [H/C]$, $i_q$ is the unique integer in $[2^C]$ with $g(i_q) = (a_{qC}, a_{qC+1}, \ldots, a_{(q+1)C-1})$. For each $h \in [H]$, we have $r_h = \epsilon$ if $h = (q+1)C - 1$ and $i_k = i_q^*$ for some $k \in [H/C]$. Otherwise, we have $r_h = 0$.

Suppose there is an RL algorithm that returns a policy $\pi$ with suboptimality less than $H/C \cdot \epsilon$, and since the value of the optimal policy is $H/C \cdot \epsilon$, we must have $r_h = \epsilon$ for some $h \in [H]$ where $(s_0, a_0, r_0, \ldots, s_{H-1}, a_{H-1}, r_{H-1})$ is the trajectory obtained by following the policy $\pi$. This implies the existence of $q \in [H/C]$ with $g(i_q^*) = (a_{qC}, a_{qC+1}, \ldots, a_{(q+1)C-1})$. Therefore, if there is an RL algorithm that returns a policy with suboptimality less than $H/C \cdot \epsilon$ for our hard instance, then there is an algorithm for solving $m$-INDQ$_n$ with $n = 2^C$ and $m = H/C$.

**Remark 1.** *Our construction is significantly different from Du et al. (2020). Specifically, we split a binary tree with $H$ levels into $H/C$ blocks. For each block, we show that any algorithm must incur a sample complexity of $2^C$ in order to find a policy with suboptimality less than $\epsilon$. In order to show that the overall suboptimality of the RL algorithm is $H/C \cdot \epsilon$, we develop a reduction from an intermediate problem called MULTI-INDEX-QUERY to RL, which is different from the one used in Du et al. (2020).*

## 4 Main Algorithm

In this section, we present our main algorithm that achieves the guarantee in Theorem 1.3.

---

**Algorithm 1** Elimination Algorithm for Finding the Optimal Hypotheses

---

1: **Input:** feature map $\phi$, sparsity $k$, approximation error $\epsilon$, statistical error $\epsilon_{\text{stat}}$, $\epsilon_{\text{net}}$, failure rate $\delta$

2: For each $h \in [H]$, initialize $\mathcal{P}_h = \mathcal{P}_h^0 = \{\theta : \theta_{\mathcal{M}} \in \mathcal{N}^k, |\mathcal{M}| = k, \mathcal{M} \subseteq [d]\}$, where $\mathcal{N}^k$ is the maximal $\epsilon_{\text{net}}/2$-separated subset of the Euclidean sphere $\mathbb{S}^k$.

3: Calculate $m = \frac{16k \ln((1+4/\epsilon_{\text{net}})d) + 16 \ln(H/\delta)}{\epsilon_{\text{stat}}^2}$.

4: **for** iteration $t = 0, 1, 2, \ldots$ **do**

5:     Choose $\theta_0^t = \arg\max_{\theta \in \mathcal{P}_0} V_\theta(s_0)$.

6:     **for** $h = 1, 2, \ldots, H-1$ **do**

7:         Define a policy $\pi_h^t$, where $\pi_h^t(s) = \pi_{\theta_{h'}^t}(s)$ if $s \in \mathcal{S}_{h'}$ with $h' < h$, and arbitrary otherwise. Collect $m$ trajectories following $\pi_h^t$ as a dataset

$$\mathcal{D}_h^t = \{(s_0^i, a_0^i, r_0^i, \ldots, s_{H-1}^i, a_{H-1}^i, r_{H-1}^i)\}_{i \in [m]}.$$

8:         Choose $\theta_h^t = \arg\max_{\theta \in \mathcal{P}_h} \sum_{i \in [m]} V_\theta(s_h^i)$, where $s_h^i$ are from dataset $\mathcal{D}_h^t$.

9:     **end for**

10:     Collect $m$ trajectories following a policy $\pi^t = \pi_{\theta^t}$ as a dataset

$$\mathcal{D}_H^t = \{(s_0^i, a_0^i, r_0^i, \ldots, s_{H-1}^i, a_{H-1}^i, r_{H-1}^i)\}_{i \in [m]}.$$

11:     For each $h \in [H]$, calculate using dataset $\mathcal{D}_H^t$:

$$\hat{\mathcal{E}}_h^t = \begin{cases} \frac{1}{m} \sum_{i=1}^m \left( \langle \phi(s_h^i, a_h^i), \theta_h^t \rangle - r_h^i - V_{\theta_{h+1}^t}(s_{h+1}^i) \right), & \text{if } h \in [H-1] \\ \frac{1}{m} \sum_{i=1}^m \left( \langle \phi(s_{H-1}^i, a_{H-1}^i), \theta_{H-1}^t \rangle - r_{H-1}^i \right), & \text{if } h = H-1. \end{cases}$$

12:     **if** $\hat{\mathcal{E}}_h^t \le 2\epsilon + 2\epsilon_{\text{net}} + 3\epsilon_{\text{stat}}$ for each $h \in [H-1]$, and $\hat{\mathcal{E}}_{H-1}^t \le \epsilon + \epsilon_{\text{net}} + \epsilon_{\text{stat}}$ **then**

13:         Terminate and output $\pi_{\theta^t}$.

14:     **else**

15:         Update $\mathcal{P}_h = \mathcal{P}_h \backslash \{\theta_h^t\}$, for all $h \in [H-1]$ satisfying $\hat{\mathcal{E}}_h^t > 2\epsilon + 2\epsilon_{\text{net}} + 3\epsilon_{\text{stat}}$, or $h = H-1$ satisfying $\hat{\mathcal{E}}_{H-1}^t > \epsilon + \epsilon_{\text{net}} + \epsilon_{\text{stat}}$.

16:     **end if**

17: **end for**

---

**Overview.** Here we give an overview of the design of Algorithm 1. We remark that, by Assumption 1, each horizon $h$ has a different optimal $\theta_h$. Therefore, a brute-force algorithm would have a sample complexity with exponential dependency on $H$, while our algorithm has a polynomial dependency on $H$.

First, we approximate all candidate parameter $\theta$ with a finite set by creating a maximal $\epsilon_{\text{net}}/2$-separated subset of the euclidean sphere $\mathbb{S}^{k-1}$, denoted by $\mathcal{N}^k$, and a set of all $k$-sized subset of $[d]$. Then, for each $h \in [H]$, we maintain a set of parameter candidates $\mathcal{P}_h$. Initially, $\mathcal{P}_h$ is set to be all parameters approximated by $\mathcal{N}^k$ and $k$-sized subset of $[d]$, i.e. $\mathcal{P}_h^0 = \{\theta : \theta_{\mathcal{M}} \in \mathbb{S}^k, |\mathcal{M}| = k, \mathcal{M} \subseteq [d]\}$ where $\theta_{\mathcal{M}}$ is the $k$-dimension sub-vector of $\theta$ with indices corresponding to $\mathcal{M}$. The set $\mathcal{P}_h^0$ is then finite for all $h \in [H]$: $|\mathcal{P}_h^0| \le (1 + 4/\epsilon_{\text{net}})^k \cdot \binom{d}{k}$ (Dong & Yang, 2023).

During the execution of Algorithm 1, for all $h \in [H]$, we eliminate parameter candidates $\theta$ from $\mathcal{P}_h$ if we are certain that $\theta \ne \hat{\theta}_h^*$, where $\hat{\theta}^* = (\hat{\theta}_0^*, \hat{\theta}_1^*, \ldots, \hat{\theta}_{H-1}^*)$ is a sequence of parameters that is in $\mathcal{P}_h^0$ and is closest to the $\theta^*$ that satisfies Assumption 1, i.e. $\hat{\theta}_h^* = \arg\min_{\theta \in \mathcal{P}_h^0} \|\theta_h^* - \theta\|$. Therefore, in Algorithm 1, we only consider $\theta = (\theta_0, \theta_1, \ldots, \theta_{H-1})$ if $\theta_h \in \mathcal{P}_h$ for all $h \in [H]$. In the $t$-th iteration, we choose a parameter $\theta^t = (\theta_0^t, \theta_1^t, \ldots, \theta_{H-1}^t)$ so that $\theta_h^t$ maximizes $\mathbb{E}[V_{\theta_h^t}(s_h)]$ and $\theta_h^t \in \mathcal{P}_h$ for all $h \in [H]$. We then collect $m$ trajectories to form a dataset $\mathcal{D}_H^t$ by following the

policy induced by $\theta^t$. Based on $\mathcal{D}_H^t$, we calculate the empirical Bellman error $\hat{\mathcal{E}}_h^t$ for each $h \in [H]$, which is the empirical estimate of the average Bellman error defined as follows.

**Definition 3** (Average Bellman error). *For a sequence of parameters $\theta^t = (\theta_0^t, \theta_1^t, \ldots, \theta_{H-1}^t)$, the average Bellman error of $\theta^t$ is defined as $\mathcal{E}_h^t = \mathbb{E}[\langle \phi(s_h, a_h), \theta_h^t \rangle - r(s_h, a_h) - V_{f_{h+1}^t}(s_{h+1})]$ when $h \in [H-1]$ and $\mathcal{E}_{H-1}^t = \mathbb{E}[\langle \phi(s_{H-1}, a_{H-1}), \theta_{H-1}^t \rangle - r(s_{H-1}, a_{H-1})]$ for level $H-1$. Here, $(s_0, a_0, r_0, \ldots, s_{H-1}, a_{H-1}, r_{H-1})$ is a trajectory obtained by following $\pi_{\theta^t}$.*

Intuitively, the Bellman error at level $h$ measures the consistency of $\theta_h^t$ and $\theta_{h+1}^t$ for the state-action distribution induced by $\pi_{\theta^t}$. In each iteration of Algorithm 1, we check if $\hat{\mathcal{E}}_h^t$ is small for all $h \in [H]$. If so, the algorithm terminates and returns the policy $\pi_{\theta^t}$. Otherwise, for all levels $h \in [H]$ where $\hat{\mathcal{E}}_h^t$ is large, we eliminate $\theta_h^t$ from $\mathcal{P}_h$ and proceed to the next iteration.

Now we give the analysis of Algorithm 1.

**Sample Complexity.** To bound the sample complexity of Algorithm 1, it suffices to give an upper bound on the number of iterations, since in each iteration, the number of trajectories sampled by the algorithm is simply $Hm = 16H(k\ln((1 + 4/\epsilon_{\text{net}})d) + \ln(H/\delta))/(\epsilon_{\text{stat}}^2)$. The following lemma gives an upper bound on the number of iterations of Algorithm 1. The proof is given by counting the number of parameters in the parameter space. The detailed proof is given in Appendix B.1.

**Lemma 4.1.** *For any MDP instance with horizon $H$ and satisfying Assumption 1 with sparsity $k$, Algorithm 1 runs for at most $(1 + 4/\epsilon_{\text{net}})^k \binom{d}{k} H$ iterations.*

**Remark 2.** *Previous works (Weisz et al., 2022; Wang et al., 2021b) show that even when the optimal Q-function is well-specified, any RL algorithm would require a sample size with exponential dependency on $d$ or $H$. Note that this is equivalent to the case where the sparsity $k = d$ and the approximation error $\epsilon = 0$ in our setting. Therefore, in our misspecified setting, which is strictly harder, exponential dependency on $k$ is unavoidable, unless we can accept an exponential dependency on $H$.*

**Suboptimality of the Returned Policy.** We now show that with probability at least $1 - \delta$, the suboptimality of the returned policy is at most $(2\epsilon + 2\epsilon_{\text{net}} + 4\epsilon_{\text{stat}})H$. First, we define a high probability event $E$, which we will condition on in the remaining part of the analysis.

**Definition 4.** *Define $E$ as the event that $|\mathcal{E}_h^t - \hat{\mathcal{E}}_h^t| \leq \epsilon_{\text{stat}}$ and $|\mathbb{E}_{s_h \sim \pi_h^t} V_\theta(s_h) - \sum_{i \in [m]} V_\theta(s_h^i)| \leq \epsilon_{\text{stat}}$ (where $s_h^i$ is from $\mathcal{D}_h^t$) for all iterations $t$, horizon $h \in [H]$, and parameter $\theta \in \mathcal{P}_h^0$.*

**Lemma 4.2.** *Event $E$ holds with probability at least $1 - \delta$.*

To prove Lemma 4.2, we first consider a fixed level $h$ and iteration $t$. Since the empirical Bellmen error $\hat{\mathcal{E}}_h^t$ is simply the empirical estimate of $\mathcal{E}_h^t$, and $\sum_{i \in [m]} V_\theta(s_h^i)$ is simply an empirical estimate of $\mathbb{E}_{s_h \sim \pi_h^t} V_\theta(s_h)$, applying the Chernoff-Hoeffding inequality respectively would suffice. Moreover, the number of iterations has an upper bound given by Lemma 4.1. Therefore, Lemma 4.2 follows by applying a union bound over all $h \in [H]$, $t \in [(1 + 4/\epsilon_{\text{net}})^k \binom{d}{k} H]$ and parameter $\theta \in \mathcal{P}_h^0$.

We next show that, conditioned on event $E$ defined above, for the sequence of parameters $\theta^* = (\theta_0^*, \theta_1^*, \ldots, \theta_{H-1}^*)$ that satisfies Assumption 1, we never eliminate $\hat{\theta}_h^*$ from $\mathcal{P}_h$, for all $h \in [H]$.

**Lemma 4.3.** *Conditioned on event $E$ defined in Definition 4, for a sequence of parameters $(\theta_0^*, \theta_1^*, \ldots, \theta_{H-1}^*)$ that satisfies Assumption 1, and their approximations $\hat{\theta}_h^* = \arg\min_{\theta \in \mathcal{P}_h^0} \|\theta_h^* - \theta\|$, during the execution of Algorithm 1, $\hat{\theta}_h^*$ is never eliminated from $\mathcal{P}_h$ for all $h \in [H]$.*

To prove Lemma 4.3, the main observation is that, for $h \in [H-1]$ the average Bellman error induced by $\hat{\theta}_h^*$ and $\theta_{h+1}^t = \arg\max_{\theta \in \mathcal{P}_{h+1}} \mathbb{E}_{s_{h+1}}[V_\theta(s_{h+1})]$ is always upper bounded by $2(\epsilon + \epsilon_{\text{net}})$, regardless of the distribution of $(s_h, a_h)$ (cf. Definition 3). Conditioned on event $E$, the empirical Bellman error induced by $\hat{\theta}_h^*$ and $\theta_{h+1}^t$ is at most $2\epsilon + 2\epsilon_{\text{net}} + 3\epsilon_{\text{stat}}$. Similarly, the empirical Bellman error induced by $\hat{\theta}_{H-1}^*$ is at most $\epsilon + \epsilon_{\text{net}} + \epsilon_{\text{stat}}$. In Algorithm 1, we eliminate function $\theta_h^t$ only when the empirical Bellman error is larger than these (Line 15). Thus, $\hat{\theta}_h^*$ is never eliminated.

We now show the suboptimality of the policy returned by Algorithm 1 is at most $(2\epsilon + 2\epsilon_{\text{net}} + 4\epsilon_{\text{stat}})H$.

**Lemma 4.4.** *For any MDP instance satisfying Assumption 1, conditioned on event $E$ defined in Definition 4, Algorithm 1 returns a policy $\pi$ satisfying $V^* - V^\pi \leq (2\epsilon + 2\epsilon_{\text{net}} + 4\epsilon_{\text{stat}})H$.*

To prove Lemma 4.4, we first recall the policy loss decomposition lemma (Lemma 1 in Jiang et al. (2017)), which states that for a policy induced by a sequence of parameters $\theta = (\theta_0, \theta_1, \ldots, \theta_{H-1})$, $V_{\theta_0}(s_0) - V^{\pi_\theta}$ is upper bounded by the summation of average Bellman error over all levels $h \in [H]$. When Algorithm 1 terminates, the empirical Bellman error must be small for all $h \in [H]$, and therefore, the average Bellman error is small by definition of the event $E$. Moreover, in Line 5 of Algorithm 1, we always choose a parameter $\theta$ that maximizes $V_\theta(s_0)$. Since the sequence of functions $\hat{\theta}^* = (\hat{\theta}_0^*, \hat{\theta}_1^*, \ldots, \hat{\theta}_{H-1}^*)$ is never eliminated by Lemma 4.3, we must have $V_{\theta_0}(s_0) \geq V_{\theta_0^*}(s_0) \geq V^* - \epsilon - \epsilon_{\text{net}}$, which gives an upper bound on the suboptimality of the policy returned by Algorithm 1. Combining Lemma 4.1, Lemma 4.2 and Lemma 4.4, we can prove Theorem 1.3.

**Remark 3.** *While Algorithm 1 assumes the sparsity constant $k$ is known, it can be easily adapted to the setting where $k$ is not known beforehand. For such a setting, we could enumerate $k$ starting from $k = 1$, and use the following observations: 1) If the true $k$, say $k^*$, is larger than $k$, then running our algorithm with sparsity $k$ will eliminate all the parameters in the parameter space $\mathcal{P}_h$ for some horizon $h$. 2) If for all $h$, there exists one parameter in $\mathcal{P}_h$ that has not been deleted, the we have identified $k^*$. The sample complexity of this process is asymptotically the same as running Algorithm 1 with known $k$, since the true $k^*$ dominates the sample complexity.*

**Implications.** We can think of the bandit setting as an MDP with $H = 1$ and derive the following.

**Corollary 4.5.** *For the bandit setting satisfying Assumption 1, Algorithm 1 returns an action $\hat{a}$ such that $r(a^*) - r(\hat{a}) \leq 2\epsilon + 2\epsilon_{\text{net}} + 4\epsilon_{\text{stat}}$.*

**Remark 4.** *Here we compare Corollary 4.5 with the result in Dong & Yang (2023). Scrutinizing the analysis in Dong & Yang (2023), the suboptimality achieved by their algorithm is $4\epsilon + \epsilon_{\text{stat}}$, which is worse than our suboptimality guarantee. On the other hand, the algorithm in Dong & Yang (2023) also returns a parameter $\theta$ such that $|\langle \phi(a), \theta \rangle - r(a)| \leq 2\epsilon + \epsilon_{\text{stat}}$ for all $a \in \mathcal{A}$ (which is the best possible according to Theorem 1.1), where our algorithm only returns a near-optimal action.*

## 5 CONCLUSION

We studied the RL problem where the optimal $Q$-functions can be approximated by a linear function with constant sparsity $k$, up to an error of $\epsilon$. We design a new algorithm with polynomial sample complexity, while the suboptimality of the returned policy is $O(H\epsilon)$, which is shown to be near-optimal by an information-theoretic hardness result.

While our algorithm achieves near-optimal suboptimality, future work could focus on improving its sample efficiency, particularly by reducing the dependence on the horizon length $H$ or eliminating the exponential dependency on $\epsilon_{\text{net}}$.

Future work could also consider achieving the $H/C \cdot \epsilon$ versus $\exp(C)$ trade-off in the upper bounds, similar to Theorem 3.2. For deterministic transition, we could segment the horizon into blocks and conduct exhaustive exploration within them. For stochastic transitions, importance sampling techniques, such as those of Kearns et al. (1999), could be explored.

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

# A  PROOFS IN SECTION 3

## A.1  PROOF OF THEOREM 3.1

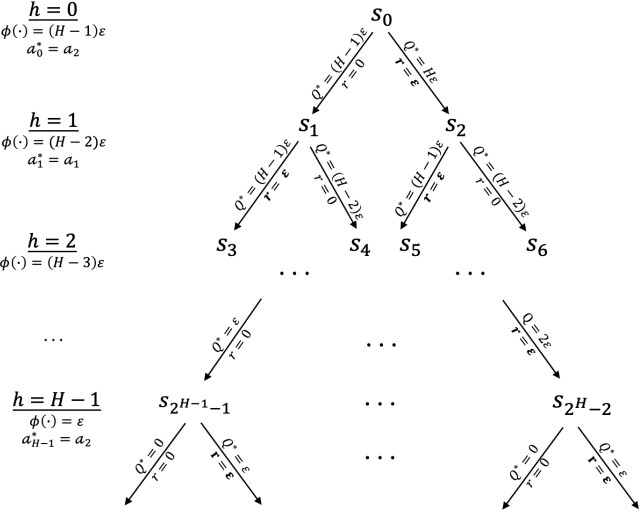

Figure 2: Illustration of the hard instance for Theorem 3.1.

*Proof.* Consider an input distribution where $a_h^*$ is drawn uniformly random from $\{a_1, a_2\}$. By Yao's minimax principle, it suffices to consider the best deterministic algorithm, say $A$. Note that, since we have no sampling ability, a deterministic algorithm in this setting can be seen as a function that takes in feature function $\phi$ and returns a policy $\pi$. Also, for all instances supported by this distribution, their inputs $\phi$ are the same. Thus, the policy returned by $A$ is fixed. Denote the policy as $\pi$, and denote the trajectory following $\pi$ as $(s_0, a_0, r_0, s_1, a_1, r_1, \ldots, s_{H-1}, a_{H-1}, r_{h-1})$. The suboptimality of $\pi$ can be written as

$$V^* - V^\pi = \sum_{h=0}^{H-1} \epsilon \cdot \mathbb{I}[a_h^* \neq a_h]$$

Since $a_h$ is fixed and $a_h^*$ is drawn uniformly random from $\{a_1, a_2\}$, $\mathbb{I}[a_h^* \neq a_h] = 1$ with probability $1/2$. Thus, $(V^* - V^\pi)/\epsilon$ is a binomial random variable, or $(V^* - V^\pi)/\epsilon \sim B(H, 1/2)$. The expectation of $(V^* - V^\pi)$ is then $H\epsilon/2$, and its variance is $H\epsilon^2/4$. Using Chebyshev inequality, with probability 0.99, we have

$$V^* - V^\pi \geq \frac{1}{2}H\epsilon - 5\epsilon\sqrt{H} = \Omega(H\epsilon)$$

for sufficiently large $H \geq 100$. $\qquad\square$

## A.2  PROOF OF LEMMA 3.3

*Proof.* Consider an input distribution where $i^* = (i_0^*, i_2^*, \ldots, i_{m-1}^*)$ is drawn uniformly random from $[n]^m$. Let $c(i^*, a)$ be the query complexity of running algorithm $a$ to solve the problem with correct indices $i^*$. Assume there exists a 0.1-correct algorithm $\mathcal{A}$ for $m$-INDQ$_n$ that queries less than $0.9n$ times in the worst case. Then, using Yao's minimax principle, there exists a deterministic algorithm $\mathcal{A}'$ with $c(i^*, \mathcal{A}') < 0.9n$ for all $i^* \in [n]^m$, such that

$$\mathbb{P}[\mathcal{A}' \text{outputs } (j, i_j^*) \text{ for some } j \in [m]] \geq 0.9.$$

We may assume that the sequence of queries made by $\mathcal{A}'$ is fixed until it correctly guesses one of $i_j^*$. This is because $\mathcal{A}'$ is deterministic, and the responses $\mathcal{A}'$ receives are the same (i.e. all guesses are incorrect) until it correctly queries $(j, i_j^*)$ for some $j$. Let $S = \{s_1, \ldots, s_k\}$ be the sequence of first $k$ guesses made by $\mathcal{A}'$, and let $I_{BAD} \subset [n]^m$ be a set of all possible $i^*$'s such that the guesses in $S$

are all incorrect. Denote the number of guesses on $\text{INDQ}_n^{(j)}$ in $S$ by $n_j$, then $n_j$'s are also fixed, and $\sum_{j \in [m]} n_j = k$. The size of $I_{BAD}$ then satisfies

$$|I_{BAD}| = \Pi_{j=0}^{m-1}(n - n_j) \geq (n - k)n^{m-1}$$

Set $k$ as the worst-case query complexity of $\mathcal{A}'$. Then, for all $i^* \in I_{BAD}$, the output of $\mathcal{A}'$ is incorrect. Since $i^*$ is drawn uniformly random from $[n]^m$, the probability of $\mathcal{A}'$ being incorrect is

$$\mathbb{P}[\mathcal{A}' \text{ is incorrect}] = \frac{|I_{BAD}|}{|[n]^m|} \geq \frac{(n - k)n^{m-1}}{n^m} > \frac{(n - 0.9n)n^{m-1}}{n^m} > 0.1,$$

where in the second to last inequality we used $k < 0.9n$.

However, this contradicts with the fact that $\mathbb{P}[\mathcal{A}' \text{outputs } (j, i_j^*) \text{ for some } j \in [m]] \geq 0.9$. Thus, there does not exist a $0.1$-correct algorithm that solves the problem with less than $0.9n$ queries in the worst case. □

### A.3  Proof of Theorem 3.2

*Proof.* First, we prove our claim based on the assumption that $C$ is an integer that divides $H$. We can create the hard instance described in Section 3.2.

We reduce the problem to $H/C$-$\text{INDQ}_{2^C}$. Assume there exists an algorithm $\mathcal{A}$ that takes less than $0.9 \cdot 2^C \cdot C$ samples, such that, with probability at least $0.9$, it outputs a policy $\pi$ with suboptimality $V^* - V^\pi < H/C \cdot \epsilon$. By definition, at round $i$, $\mathcal{A}$ interacts with the MDP instance by following a trajectory $(s_0, a_0^i, r_0^i, ..., s_{H-1}^i, a_{H-1}^i, r_{H-1}^i)$. Based on $\mathcal{A}$, we create an algorithm $\mathcal{A}'$ for $H/C$-$\text{INDQ}_{2^C}$ as follows. Consider $\mathcal{A}$ is querying the trajectory $(s_0, a_0^i, r_0^i, ..., s_{H-1}^i, a_{H-1}^i, r_{H-1}^i)$. For each $q \in \{0, \ldots, H/C - 1\}$, we can map $(a_{qC}^i, \ldots, a_{(q+1)C-1}^i)$ to an index in $[2^C]$ using the bijection $g$. Thus, we make a sequence of $H/C$ guesses, $\{(q, g(a_{qC}^i, \ldots, a_{(q+1)C-1}^i))\}_{q=0}^{H/C-1}$, to the $H/C$-$\text{INDQ}_{2^C}$. If the guess $(q, g(a_{qC}^i, \ldots, a_{(q+1)C-1}^i))$ is correct for some $q$, $\mathcal{A}$ receives a reward of $\epsilon$ at level $(q+1)C - 1$, i.e. $r_{(q+1)C-1}^i = r(s_{(q+1)C-1}^i, a_{(q+1)C-1}^i) = \epsilon$. For all other state-action pairs in the trajectory, algorithm $\mathcal{A}$ receives zero reward. Since $\mathcal{A}$ takes less than $0.9 \cdot 2^C \cdot C$ samples, it queries less than $0.9 \cdot 2^C \cdot C/H$ trajectories, corresponding $0.9 \cdot 2^C$ guesses to $H/C$-$\text{INDQ}_{2^C}$ in total. Recall that $\mathcal{A}$ outputs a policy $\pi$ with suboptimality $V^* - V^\pi < H/C \cdot \epsilon$ with probability at least $0.9$. This means the sequence of guesses to $H/C$-$\text{INDQ}_{2^C}$ made by $\pi$ must have at least one of them being correct. Thus, $\mathcal{A}'$ is a $0.1$-correct algorithm that solves $H/C$-$\text{INDQ}_{2^C}$ with less than $0.9 \cdot 2^C$ guesses. However, by Lemma 3.3, such an algorithm does not exist, so $\mathcal{A}$ does not exist. We conclude that any algorithm that returns a policy with suboptimality less than $H/C \cdot \epsilon$ with probability at least $0.9$ needs to sample at least $0.9 \cdot C \cdot 2^C$ times.

Now we consider when $C$ is not an integer that divides $H$. There are two cases. First, consider $C$ as an integer that does not divide $H$. Let $H' = \lfloor H/C \rfloor \cdot C$, then we can make the same construction as above for the first $H'$ horizons, and set the reward as zero for all the state-action pairs in the remaining $H - H'$ levels. Because the rewards are the same for levels $H'$ through $H - 1$, different values of $\{\pi_{H'}, \ldots, \pi_{H-1}\}$ do not make a difference to $V^\pi$. Therefore, we only care about the first $H'$ levels, so we can conclude from our above analysis that, any algorithm that returns a policy with suboptimality less than $H'/C \cdot \epsilon = \lfloor H/C \rfloor \cdot \epsilon$ with probability at least $0.9$ needs to sample at least $0.9 \cdot C \cdot 2^C$ times. For the second case, we consider when $C$ is not an integer. Let $C' = \lfloor C \rfloor$, we can apply our conclusion from the previous case. That is, any algorithm that returns a policy with suboptimality less than $\lfloor H/C' \rfloor \cdot \epsilon$ with probability at least $0.9$ needs to sample at least $0.9 \cdot C' \cdot 2^{C'}$ times. Since $2C \leq H$, we have $\lfloor H/C' \rfloor \cdot \epsilon \geq \lfloor H/C \rfloor \cdot \epsilon \geq \frac{H\epsilon}{2C}$. Also observing that $0.9 \cdot C' \cdot 2^{C'} \geq 0.1 \cdot C \cdot 2^C$, we finish the proof. □

# B  PROOFS IN SECTION 4

## B.1  PROOF OF LEMMA 4.1

*Proof.* In each iteration, we either output a policy or delete at least one function in $\mathcal{P}_h$ for some $h \in [H-1]$. Since there are $\sum_{h \in [H-1]} \left( |\mathbb{S}^k| \times \binom{d}{k} \right) \leq (1 + 4/\epsilon_{\text{net}})^k \binom{d}{k} H$ functions in total initially, the algorithm is guaranteed to terminate within $(1 + 4/\epsilon_{\text{net}})^k \binom{d}{k} H$ iterations. $\qquad\square$

## B.2  PROOF OF LEMMA 4.2

**Lemma B.1** (Deviation bound for $\mathcal{E}_h$). *For fixed iteration $t$ and horizon $h \in [H]$, with probability at least $1 - \delta'$, we have*

$$|\mathcal{E}_h^t - \hat{\mathcal{E}}_h^t| \leq 4\sqrt{\frac{\ln 2 - \ln \delta'}{2m}}.$$

*Hence, we can set $m > \frac{16(\ln 2 - \ln(\delta'))}{2\epsilon_{\text{stat}}^2}$ to guarantee that $|\mathcal{E}_h^t - \hat{\mathcal{E}}_h^t| < \epsilon_{\text{stat}}$*

*Proof.* Recall that the batch dataset $\mathcal{D}_t = \{(a_0^i, r_0^i, s_1^i, ..., a_{H-1}^i, r_{H-1}^i)\}_{i=1}^m$ is collected by playing policy $\pi_{\theta^t}$. We define $\hat{\mathcal{E}}_h^{t,i} = \langle \phi(s_{h-1}^i, a_h^i), \theta_h^t \rangle - r(s_h^i, a_h^i) - V_{\theta_{h+1}^t}(s_{h+1}^i)$, then $\hat{\mathcal{E}}_h^t = \frac{1}{m} \sum_{i=1}^m \hat{\mathcal{E}}_h^{t,i}$. By definition of $\mathcal{E}_h^t$, it satisfies

$$\mathcal{E}_h^t = \mathbb{E}[\hat{\mathcal{E}}_h^{t,i}].$$

Further, since $\langle \phi(s,a), \theta^t \rangle \in [-1, 1]$ and $r(s,a) \in [0, 1]$ for any state-action pair $(s, a)$, we have $\hat{\mathcal{E}}_h^{t,i} \in [-3, 1]$. Thus, using Chernoff-Hoeffding inequality, we get, with probability $1 - \delta'$,

$$|\mathcal{E}_h^t - \hat{\mathcal{E}}_h^t| = \left| \frac{1}{m} \sum_{i=1}^m \left( \hat{\mathcal{E}}_h^{t,i} - \mathbb{E}[\hat{\mathcal{E}}_h^t] \right) \right| \leq 4\sqrt{\frac{\ln 2 - \ln \delta'}{2m}}.$$

$\qquad\square$

**Lemma B.2** (Deviation bound for $\mathbb{E}_{s_h \sim \pi_h^t} V_\theta(s_h)$). *For fixed iteration $t$, horizon $h \in [H]$, and parameter $\theta \in \mathcal{P}_h$, with probability at least $1 - \delta'$, we have*

$$\left| \mathbb{E}_{s_h \sim \pi_h^t} V_\theta(s_h) - \frac{1}{m} \sum_{i \in [m]} V_\theta(s_h^i) \right| \leq \sqrt{\frac{\ln 2 - \ln \delta'}{2m}}.$$

*Hence, we can set $m > \frac{\ln 2 - \ln(\delta')}{2\epsilon_{\text{stat}}^2}$ to guarantee that $|\mathcal{E}_h^t - \hat{\mathcal{E}}_h^t| < \epsilon_{\text{stat}}$*

*Proof.* Recall that the batch dataset $\mathcal{D}_h^t = \{(s_0^i, a_0^i, r_0^i, \ldots, s_{h-1}^i, a_{h-1}^i, r_{h-1}^i, s_h^i)\}_{i=1}^m$ is collected by playing policy $\pi_h^t$. By definition, it satisfies

$$\mathbb{E}_{s_h \sim \pi_h^t} V_\theta(s_h) = \mathbb{E}_{\mathcal{D}_h^t} \left[ \frac{1}{m} \sum_{i \in [m]} V_\theta(s_h^i) \right].$$

Further, since $V_\theta(s) \in [0, 1]$ for any state $s$, using Chernoff-Hoeffding inequality, we have with probability $1 - \delta'$,

$$\left| \mathbb{E}_{s_h \sim \pi_h^t} V_\theta(s_h) - \frac{1}{m} \sum_{i \in [m]} V_\theta(s_h^i) \right| \leq \sqrt{\frac{\ln 2 - \ln \delta'}{2m}}.$$

$\qquad\square$

*Proof.* Define $E_{t,h}^{\mathcal{E}}$ to be the event

$$E_{t,h}^{\mathcal{E}} = \{|\mathcal{E}_h^t - \hat{\mathcal{E}}_h^t| \le \epsilon_{\text{stat}}\},$$

then by Lemma B.1, $\mathbb{P}(E_{t,h}^{\mathcal{E}}) \ge 1 - \delta/(2(1 + 4/\epsilon_{\text{net}})^{2k}\binom{d}{k}^2 H^2)$ for all iterations $t \in [(1 + 4/\epsilon_{\text{net}})^k \binom{d}{k}H]$ and horizon $h \in [H]$.

Define $E_{t,h,\theta}^V$ to be the event

$$E_{t,h,\theta}^V = \left\{ \left| \mathbb{E}_{s_h \sim \pi_h^t} V_\theta(s_h) - \frac{1}{m} \sum_{i \in [m]} V_\theta(s_h^i) \right| \le \epsilon_{\text{stat}} \right\},$$

then by Lemma B.2, $\mathbb{P}(E_{t,h,f}^V) \ge 1 - \delta/(2(1 + 4/\epsilon_{\text{net}})^{2k}\binom{d}{k}^2 H^2)$ for all iterations $t \in [(1 + 4/\epsilon_{\text{net}})^k \binom{d}{k}H]$, horizon $h \in [H]$, and $\theta \in \mathcal{P}_h$.

We can lower bound the probability of $E$ by union bound

$$\mathbb{P}(E) \ge 1 - \sum_t \sum_{h \in [H]} \mathbb{P}(\bar{E}_{t,h}^{\mathcal{E}}) - \sum_t \sum_{h \in [H]} \sum_{\theta \in \mathcal{P}_h} \mathbb{P}(\bar{E}_{t,h,\theta}^V) \ge 1 - \delta.$$

$\square$

### B.3 PROOF OF LEMMA 4.3

*Proof.* Let $\hat{\theta}^* = (\hat{\theta}_0^*, \dots, \hat{\theta}_{H-1}^*)$ be the sequence of parameters such that, for each $h \in [H]$, the non-zero sub-vector of $\hat{\theta}_h^*$ is in $\mathcal{N}^k$ and is closest to the non-zero indices in $\theta^*$. Then, since $\mathcal{N}^k$ is $\epsilon_{\text{net}}/2$-maximal, we have by Assumption 1 that

$$|\langle \phi(s,a), \hat{\theta}_h^* \rangle - Q^*(s,a)| \le |\langle \phi(s,a), \theta_h^* \rangle - Q^*(s,a)| + |\langle \phi(s,a), \hat{\theta}_h^* \rangle - \langle \phi(s,a), \theta_h^* \rangle| \le \epsilon + \epsilon_{\text{net}},$$

for all $s$ in horizon $h$ and action $a \in \mathcal{A}$.

At iteration $t$, algorithm 1 deletes $\hat{\theta}_h^*$ if and only if one of the following two cases happens: (1) $h < H - 1$, $\theta_h^t = \hat{\theta}_h^*$, and $\hat{\mathcal{E}}_h^t > 2\epsilon + 2\epsilon_{\text{net}} + 3\epsilon_{\text{stat}}$, (2) $h = H - 1$, $\theta_{H-1}^t = \hat{\theta}_{h+1}^*$ and $\hat{\mathcal{E}}_{H-1}^t > \epsilon + \epsilon_{\text{net}} + \epsilon_{\text{stat}}$.

For any state-action pair $(s_h, a_h)$ at level $h$ where $h \in [H-1]$, we observe by definition that

$$Q^*(s_h, a_h) - \mathbb{E}[r(s_h, a_h)] - \mathbb{E}[V_{h+1}^*(s_{h+1})] = 0.$$

Thus, we can upper bound $\mathcal{E}_h^t$ by

$$\begin{aligned} \mathcal{E}_h^t &= \mathbb{E}[\langle \phi(s_h, a_h), \hat{\theta}_h^* \rangle - r(s_h, a_h) - V_{\theta_{h+1}}(s_{h+1})] \\ &\le \mathbb{E}[(Q^*(s_h, a_h) + \epsilon + \epsilon_{\text{net}}) - r(s_h, a_h) - V_{\theta_{h+1}}(s_{h+1})] \end{aligned} \qquad \text{(By Assumption 1)}$$

Here, $(s_0, a_0, r_0, \dots, s_h, a_h, r_h)$ is a trajectory following $\pi_{\theta^t}$, and $s_{h+1} \sim P(s_h, a_h)$.

Recall that $\theta_h^t$ is chosen by taking the function that gives maximum empirical value at level $h$, so

$$\frac{1}{m} \sum_{i \in [m]} V_{\theta_h^t}(s_h^i) \ge \frac{1}{m} \sum_{i \in [m]} V_{\hat{\theta}_h^*}(s_h^i),$$

where $s_h^i$ are taken from the dataset $\mathcal{D}_h^t$. Moreover, we are conditioned under event $E$, so we have

$$\mathbb{E}[V_{\theta_h^*}(s_h)] - \mathbb{E}[V_{\theta_h^t}(s_h)] \le \left( \frac{1}{m} \sum_{i \in [m]} V_{\theta_h^t}(s_h^i) + \epsilon_{\text{stat}} \right) - \left( \frac{1}{m} \sum_{i \in [m]} V_{\theta_h^*}(s_h^i) - \epsilon_{\text{stat}} \right) \le 2\epsilon_{\text{stat}}$$

for all $h$ and $t$.

For the first case, we consider $h \in [H-1]$. We have

$$
\begin{aligned}
\mathcal{E}_h^t \leq & \mathbb{E}[(Q^*(s_h, a_h) + \epsilon + \epsilon_{\text{net}}) - r(s_h, a_h) - V_{\theta_{h+1}^t}(s_{h+1})] \\
\leq & \mathbb{E}[Q^*(s_h, a_h) - r(s_h, a_h) - (V_{f_{h+1}^*}(s_{h+1}) - 2\epsilon_{\text{stat}})] + \epsilon + \epsilon_{\text{net}} \\
= & \mathbb{E}[Q^*(s_h, a_h) - r(s_h, a_h) - f_{h+1}^*(s_{h+1}, \pi_{h+1}^*(s_{h+1}))] + \epsilon + \epsilon_{\text{net}} + 2\epsilon_{\text{stat}} \\
& \hspace{3cm} (\text{since } V_{f_{h+1}^*} = \max_{a \in \mathcal{A}} f_{h+1}^*(s_{h+1}, a)) \\
\leq & \mathbb{E}[Q^*(s_h, a_h) - r(s_h, a_h) - (Q^*(s_{h+1}, \pi_{h+1}^*(s_{h+1})) - \epsilon - \epsilon_{\text{net}})] + \epsilon + \epsilon_{\text{net}} + 2\epsilon_{\text{stat}} \\
& \hspace{3cm} (\text{By Assumption 1}) \\
= & \mathbb{E}[Q^*(s_h, a_h) - r(s_h, a_h) - Q^*(s_{h+1}, \pi_h^*(s_{h+1}))] + 2\epsilon + 2\epsilon_{\text{net}} + 2\epsilon_{\text{stat}} \\
= & 2\epsilon + 2\epsilon_{\text{net}} + 2\epsilon_{\text{stat}}. \hspace{2cm} (\text{since } Q^*(s_{h+1}, \pi_{h+1}^*(s_{h+1})) = V_{h+1}^*(s_{h+1}))
\end{aligned}
$$

Given that we are conditioned under event $E$, $\hat{\mathcal{E}}_h^t - \mathcal{E}_h^t \leq \epsilon_{\text{stat}}$ for all iteration $t$ and all horizon $h$. Thus, $\hat{\mathcal{E}}_h^t < 2\epsilon + 2\epsilon_{\text{net}} + 3\epsilon_{\text{stat}}$.

For the second case, we consider $h = H - 2$. We have

$$
\mathcal{E}_{H-1}^t \leq \mathbb{E}[(r(s_{H-1}, a_{H-1}) + \epsilon) - r(s_{H-1}, a_{H-1})] = \epsilon + \epsilon_{\text{net}},
$$

because $H - 1$ is the last level.

Again, given that we are conditioned under event $E$, we have $\hat{\mathcal{E}}_{H-1}^t - \mathcal{E}_{H-1}^t \leq \epsilon_{\text{stat}}$, so $\hat{\mathcal{E}}_{H-1}^t < \mathcal{E}_{H-1}^t + \epsilon_{\text{stat}} \leq \epsilon + \epsilon_{\text{net}} + \epsilon_{\text{stat}}$. □

## B.4 PROOF OF LEMMA 4.4

*Proof.* Algorithm 1 terminates and returns a policy at iteration $t$ only if $\theta^t$ satisfies the conditions in line 6, and by Lemma 4.3, there always exists a nice sequence of functions $\{\hat{\theta}_h^*\}_{h=0}^{H-1}$ that satisfies these conditions. Also, Lemma 4.1 indicates that the algorithm terminates within a finite number of iterations. Thus, algorithm 1 is guaranteed to terminate and return a policy.

Let the output policy be $\pi_{\theta^t}$, i.e. $\hat{\mathcal{E}}_h^t \leq 2\epsilon + 2\epsilon_{\text{net}} + 3\epsilon_{\text{stat}}$ for all $h \in [H-2]$ and $\hat{\mathcal{E}}_{H-1}^t \leq \epsilon + \epsilon_{\text{net}} + \epsilon_{\text{stat}}$. The loss of this policy can be bounded by

$$
\begin{aligned}
V^*(s_0) - V^{\pi_{\theta_t}}(s_0) = & Q^*(s_0, \pi^*(s_0)) - V^{\pi_{\theta^t}}(s_0) \\
\leq & (\langle \phi(s_0, \pi^*(s_0)), \hat{\theta}_0^* \rangle + \epsilon + \epsilon_{\text{net}}) - V^{\pi_{\theta^t}}(s_0) \hspace{1cm} (\text{By Assumption 1}) \\
\leq & (\langle \phi(s_0, \pi_{\theta_0^t}(s_0)), \theta_0^t \rangle + \epsilon + \epsilon_{\text{net}}) - \mathbb{E}[\sum_{h=0}^{H-1} r(s_h, a_h)] \\
& \hspace{2cm} (\text{since } \theta_0^t \text{ is chosen by taking the maximum}) \\
= & \epsilon + \epsilon_{\text{net}} + \mathbb{E}\Big[\sum_{h=0}^{H-1} \langle \phi(s_h, a_h), \theta_h^t \rangle - r(s_h, a_h) - \langle \phi(s_{h+1}, a_{h+1}), \theta_{h+1}^t \rangle\Big] \\
& \hspace{4cm} (\text{telescoping sum}) \\
= & \epsilon + \epsilon_{\text{net}} + \sum_{h=0}^{H-1} \mathbb{E}\Big[\langle \phi(s_h, a_h), \theta_h^t \rangle - r(s_h, a_h) - \langle \phi(s_{h+1}, a_{h+1}), \theta_{h+1}^t \rangle\Big] \\
& \hspace{4cm} (\text{linearity of expectation}) \\
= & \epsilon + \epsilon_{\text{net}} + \sum_{h=0}^{H-1} \mathcal{E}_h^t \leq \epsilon + \epsilon_{\text{net}} + \sum_{h=0}^{H-1} (\hat{\mathcal{E}}_h^t + \epsilon_{\text{stat}}) \leq (2\epsilon + 2\epsilon_{\text{net}} + 4\epsilon_{\text{stat}})H
\end{aligned}
$$

□

# C ADDITIONAL PROOFS

## C.1 PROOF OF THEOREM 1.1

*Proof.* We construct a hard instance as follows. For each $a \in [n]$, define $\phi(a) = \epsilon$. Let $\theta^*$ be randomly selected from $\{-1, 1\}$, and let $a^*$ is uniformly chosen from $\mathcal{A}$. The reward $r$ is deterministic

and is defined as

$$r(a) = \begin{cases} 2\theta^*\epsilon & \text{if } a = a^* \\ 0 & \text{otherwise.} \end{cases}$$

Therefore $|r(a) - \theta^* \cdot \phi(a)| \le \epsilon$ holds true for all actions $a \in \mathcal{A}$.

By Yao's minimax principle, it suffices to consider deterministic algorithms. Let $A$ be a deterministic algorithm that, by taking less than $0.9n$ samples, returns a $\hat{r}$ with $|\hat{r}(a) - r(a)| < 2\epsilon$ for all $a \in \mathcal{A}$ with probability at least $0.95$. We can say the sequence of actions made by $A$ is fixed until it receives a reward $r(a_t) \neq 0$ at some round $t$. This is because $A$ is deterministic, and the responses $A$ receives are the same (i.e. all actions have reward 0) until it queries $a^*$. Let $S = (a_1, \dots, a_t)$ be the sequence of actions made by $A$. Let $\mathcal{A}_{BAD} \subset \mathcal{A}$ be the set of actions that are not in S. We have

$$\begin{aligned}
\mathbb{P}[|\hat{r}(a) - r(a)| < 2\epsilon, \forall a \in \mathcal{A}] =& \mathbb{P}[|\hat{r}(a) - r(a)| < 2\epsilon, \forall a \in \mathcal{A} | a^* \in \mathcal{A}_{BAD}] \mathbb{P}[a^* \in \mathcal{A}_{BAD}] \\
& + \mathbb{P}[|\hat{r}(a) - r(a)| < 2\epsilon, \forall a \in \mathcal{A} | a^* \notin \mathcal{A}_{BAD}] \mathbb{P}[a^* \notin \mathcal{A}_{BAD}] \\
<& \mathbb{P}[|\hat{r}(a) - r(a)| < 2\epsilon, \forall a \in \mathcal{A} | a^* \in \mathcal{A}_{BAD}] \mathbb{P}[a^* \in \mathcal{A}_{BAD}] \\
& + (1 - \mathbb{P}[a^* \in \mathcal{A}_{BAD}])
\end{aligned}$$

Since $t < 0.9n$ and $a^*$ is chosen uniformly random from $\mathcal{A}$, the probability that $a^* \in \mathcal{A}_{BAD}$ is

$$\mathbb{P}[a^* \in \mathcal{A}_{BAD}] = \frac{|\mathcal{A}_{BAD}|}{|\mathcal{A}|} > \frac{1 - 0.9n}{n} = 0.1.$$

When $a^* \in \mathcal{A}_{BAD}$, the output of our deterministic algorithm must be fixed. We denote such output by $r'$. Consider a fixed $a^* \in \mathcal{A}_{BAD}$, if we have $|r'(a^*) - 2\epsilon| < 2\epsilon$, then $r'(a^*) \in (0, 4\epsilon)$, and $|r'(a^*) - (-2\epsilon))| > 2\epsilon$. Similarly, if we have $|r'(a^*) - (-2\epsilon)| < 2\epsilon$, then $|r'(a^*) - 2\epsilon| > 2\epsilon$. Since $\theta^*$ is chosen uniformly random in $\{-1, 1\}$, we know $r(a^*)$ is chosen uniformly random in $\{-2\epsilon, 2\epsilon\}$. Thus,

$$\mathbb{P}[|\hat{r}(a) - r(a)| < 2\epsilon, \forall a \in \mathcal{A} | a^* \in \mathcal{A}_{BAD}] \le \mathbb{P}[|\hat{r}(a^*) - r(a^*)| < 2\epsilon | a^* \in \mathcal{A}_{BAD}] = 0.5.$$

We have

$$\mathbb{P}[|\hat{r}(a) - r(a)| < 2\epsilon, \forall a \in \mathcal{A}] < 0.5 \cdot \mathbb{P}[a^* \in \mathcal{A}_{BAD}] + (1 - \mathbb{P}[a^* \in \mathcal{A}_{BAD}]) < 0.5 \cdot 0.1 + 0.9 = 0.95.$$

However, by our assumption on algorithm $A$, we have $\mathbb{P}[|\hat{r}(a) - r(a)| < 2\epsilon, \forall a \in \mathcal{A}] > 0.95$. $\qquad\square$

## D  BELLMAN RANK

The following definition of the general average Bellman error is helpful for our proofs in this section.

**Definition 5.** *Given any policy $\pi : \mathcal{S} \to \mathcal{A}$, feature function $\phi : \mathcal{S} \times \mathcal{A} \to \mathbb{R}^d$ and a sequence of parameters $\theta = (\theta_0, \dots, \theta_{H-1})$, the average Bellman error of $\theta$ under roll-in policy $\pi$ at level $h$ is defined as*

$$\mathcal{E}_h(\theta, \pi) = \mathbb{E}[\langle \phi(s_h, a_h), \theta_h \rangle - r_h - \max_{a \in \mathcal{A}} \langle \phi(s_{h+1}, a), \theta_{h+1} \rangle].$$

*Here, $(s_0, a_0, r_0, \dots, s_h, a_h, r_h)$ is a trajectory by following $\pi$, and $s_{h+1} \sim P(s_h, a_h)$.*

**Definition 6** (Bellman Rank)**.** *For a given MDP $\mathcal{M}$, we say that our parameter space $\mathcal{F} = \{\theta \in \mathbb{R}^d : \theta \text{ is } k\text{-sparse}, \|\theta\|_2 = 1\}$ has a Bellman rank of dimension $d$ if, for all $h \in [H]$, there exist functions $X_h : \mathcal{F} \to \mathbb{R}^d$ and $Y_h : \mathcal{F} \to \mathbb{R}^d$ such that for all $\theta, \theta' \in \mathcal{F}$,*

$$\mathcal{E}_h(\theta, \pi_{\theta'}) = \langle X_h(\theta), Y_h(\theta') \rangle.$$

For each $h \in [H]$, define $W_h \in \mathbb{R}^{d \times d}$ as the Bellman error matrix at level $h$, where the $i, j$-th index of $W_h$ is $\mathcal{E}_h(\theta_i, \pi_{\theta_j})$. Then, the Bellman rank of $\mathcal{F}$ is the maximum among the rank of the matrices $\{W_h\}_{h \in [H]}$.

We prove Proposition 1.2.

*Proof.* We again construct a deterministic MDP instance with binary trees. For simplicity, we assume $d$ is a power of 2, and we construct the instance with horizon $H = \log d$. Thus, we have $|\mathcal{S}| = 2^H - 1 = d - 1$ states. We also assume the sparsity is $k = 1$, so the parameter space $|\mathcal{F}| = d$. The rest details of state space, action space, and the transition kernel are exactly the same as in Section 3.1.

The reward is defined as $r(s, a_1) = r(s, a_2) = \epsilon$ for $s \in \mathcal{S}_{H-1}$, and $r(s, a) = 0$ for all other state-action pairs. Correspondingly, the $Q$-function satisfies that $Q^*(s, a) = \epsilon$ for all $(s, a) \in \mathcal{S} \times \mathcal{A}$.

For feature at horizon $h \in [H]$, for $j \geq 2^{h+1}$, we define the $j$-th index of $\phi(s, a)$ as $\phi(s, a)[j] = j\epsilon$ for all $(s, a) \in \mathcal{S}_h \times \mathcal{A}$. For $i \in [2^{h+1}]$ and $i$ is even, $\phi(s_{2^h-1+i}, a_1)[i] = \epsilon$ and $\phi(s_{2^h-1+i}, a_2)[i] = 0$. If $i \in [2^{h+1}]$ and $i$ is odd, then $\phi(s_{2^h-1+i}, a_1)[i] = 0$ and $\phi(s_{2^h-1+i}, a_2)[i] = \epsilon$. We also define $\phi(s, a)[i] = 0$ for all other state-action pairs.

Notice that, for any $h \in [H]$ and $i \in [2^{h+1}]$, we have can let $\theta$ be the one-hot vector with $i$-th index being 1, then $|\langle\phi(s, a), \theta\rangle - Q^*(s, a)| \leq \epsilon$ for all $(s, a) \in \mathcal{S}_h \times \mathcal{A}$, so our construction satisfies assumption 1.

Clearly, for each pair $(s, a) \in \mathcal{S}_{H-1} \times \mathcal{A}$, we can find $\theta = (\theta_0, \ldots, \theta_{H-1})$ such that the trajectory created by following $\pi_\theta$, denoted by $(s_0, a_0, r_0, \ldots, s_{H-1}, a_{H-1}, r_{H-1})$, satisfies $s_{H-1} = s$ and $a_{H-1} = a$.

Consider two parameter candidates $\theta, \theta'$. Let $(s_{H-1}, a_{H-1})$ be the state and action at level $H - 1$ when following $\pi_{\theta'}$. Since we are considering deterministic MDP, we can calculate the Bellman error at level $H - 1$ as follows

$$\mathcal{E}_{H-1}(\theta, \pi_{\theta'}) = \langle\phi(s_{H-1}, a_{H-1}), \theta_{H-1}\rangle - r(s_{H-1}, a_{H-1}) - \max_{a \in \mathcal{A}}\langle\phi(s_H, a), \theta_H\rangle$$

$$= \langle\phi(s_{H-1}, a_{H-1}), \theta_{H-1}\rangle - \epsilon \qquad \text{(since } H - 1 \text{ is the last level)}$$

$$= \begin{cases} 0 & \text{, if } \theta_{H-1} = \theta'_{H-1} \\ -\epsilon & \text{, if } \theta_{H-1} \neq \theta'_{H-1}. \end{cases}$$

Here, the last equality holds because, for each $\theta_{H-1}$, there is only one unique $(s, a) \in \mathcal{S}_{H-1} \times \mathcal{A}$ such that $\langle\phi(s, a), \theta_{H-1}\rangle = \epsilon$.

Thus, at level $H - 1$, a submatrix of the Bellman error matrix, $W \in \mathbb{R}^{d \times d}$, satisfies

$$W_{ij} = \mathcal{E}_{H-1}(\theta_i, \pi_{\theta_j}) = \begin{cases} 0 & \text{, if } i = j \\ -\epsilon & \text{, otherwise.} \end{cases}$$

In other words, $W = \epsilon(I - J)$ where $I$ is the identity matrix and $J$ is a $d \times d$ matrix with all 1s. Define matrix $W' = \frac{1}{\epsilon}(I - 1/(n-1)J)$, then we have

$$WW' = (I - J)(I - \frac{1}{n-1}J)$$

$$= I - \frac{1}{n-1}J - J + \frac{n}{n-1}J = I.$$

This means $W'$ is the inverse matrix of $W$, and $W$ is full rank. Thus, the Bellman rank is at least $d$. □

