# OpenReview forum: "Misspecified  $Q$-Learning with Sparse Linear Function Approximation: Tight Bounds on Approximation Error"
_ICLR.cc/2025/Conference — ICLR 2025 Poster_

### Official Review · Reviewer_FFJt · 2024-11-02

**Soundness:** 3
**Presentation:** 3
**Contribution:** 2
**Rating:** 6
**Confidence:** 3

**Summary:**

$\textbf{Background}$ In Q-learning with function approximation, it is assumed that the optimal Q-value function can be written as a linear function of the feature of each state, action pair, i.e., $Q^*(s, a) = <  \theta^*,  \psi(s, a)>$.
In the misspecified setting, the assumption is relaxed to be such that the optimal Q-value function is $\epsilon$-close to some linear function of the features, i.e., $| Q^* - <  \theta^*,  \psi(s, a)> | < \epsilon $.
Unfortunately, it has been shown that exponentially many samples are required in the misspecified setting even when the MDP only has $H=1$ step, which simplifies to the linear bandit setting.
Yet, a prior work shows that the hardness result can be circumvented in the linear bandit setting provided that the optimal secrete vector $\theta^*$ is further assumed to be $k$-sparse for some constant $k$. In such cases, the sample complexity for linear bandit has been improved to $O( (d/\epsilon)^k )$.

$\textbf{Contribution of this work}$ In this work, it has been shown results with a similar favor hold for general multi-step Q-learning as well. In particular, they give an algorithm that uses polynomially many samples and returns an $O(H \epsilon)$-optimal policy, where $H$ is the number of steps of the MDP.  Interestingly, the error $H \epsilon$ has been shown to be optimal up to polylogarithmic factors.

The main novelty of the proposed algorithm, compared to the prior work, is in how it chooses the policy to execute. In particular, the policy is chosen to maximize the so-called “empirical roll-in distribution” at all levels. After that, similar to prior work, the algorithm proceeds to compute average bellman error, and eliminate parameters that lead to large Bellman error (though the algorithm is a bit more conservative in the sense that it always eliminates at most one candidate parameter per level in every iteration).

**Strengths:**

The error guarantee of the algorithm proposed is shown to be optimal though it has a surprising linear scale with the number of steps $H$. The lower bound argument is quite interesting. They first demonstrate a suboptimality gap assuming that the RL algorithm is given no samples using hard instances based on binary trees. They further show that the instance can be modified so that the reward-signal is exponentially sparse. Hence, the algorithm must search the space in a brute-force manner to extract useful information.

**Weaknesses:**

The algorithm is structurally similar to the algorithm from Jiang et al., 2017. Since the algorithm eliminates one parameter per level every iteration, it is almost doing a brute-force search over the entire parameter space. As a result, the sample complexity and runtime of the algorithm both scales exponentially with the sparsity parameter $k$ of the problem.

**Questions:**

Do the authors have a guess whether the exponential dependency on $k$ in the sample complexity is necessary?

---

> ### Author Response · Authors · 2024-11-23
>
> Thank you for your careful review!
>
> First, we want to emphasize that our algorithm is not brute-force. By Assumption 1.1, each horizon $h$ has a different optimal $\theta^*_h$. Therefore, a brute-force algorithm would have a sample complexity with exponential dependency on $H$ unlike our algorithm. Moreover, In the setting where the sparsity $k$ is a constant, which is the main focus of this paper, our algorithm has a polynomial dependency on $H$ and $d$, since in our algorithm we considered a much smaller net that exploits the sparsity structure. For instance, when the optimal Q-funcion is a $d=100$ dimensional linear function, but can be approximated by a $k=2$-sparse linear function, the sample complexity and running time of our algorithm would be quadratic.
>
> **The exponential dependency on k is necessary**. Previous works (Weisz et al., 2022, Wang et al., 2021) show that even when the optimal Q-function is well-specified, any RL algorithm would have exponential dependency on $d$ or $H$. Note that this is equivalent to the case where the sparsity $k = d$ and the approximation error $\epsilon = 0$ in our setting. Therefore, in our misspecified setting which is strictly harder, exponential dependency on $k$ is unavoidable, unless we can accept an exponential dependency on $H$.
>
> References:
>
> Gellert Weisz, Csaba Szepesvari, and Andras Gyorgy. TensorPlan and the Few Actions Lower Bound for Planning in MDPs under Linear Realizability of Optimal Value Functions. 2022.
>
> Yuanhao Wang, Ruosong Wang, and Sham M. Kakade. An Exponential Lower Bound for Linearly-Realizable MDPs with Constant Suboptimality Gap. 2021

---

### Official Review · Reviewer_AMgy · 2024-11-02

**Soundness:** 3
**Presentation:** 2
**Contribution:** 2
**Rating:** 6
**Confidence:** 3

**Summary:**

In this work, the authors study RL where Q-function is sparse codable up to $\epsilon$ error.
Their main contributions are
i) showing the existence of a hard instance where achieving suboptimality less than $H \epsilon / C$ requires sample complexity exponential to C,
ii) establishing a theoretical method achieving suboptimality of $2H \epsilon + o(1)$ with polynomial sample.

**Strengths:**

- An interesting work on the RL learnability under limited knowledge of environment.
- The theoretical results are strong in the sense that almost-matching upper and lower bounds are presented.

**Weaknesses:**

- The sample complexity grows exponentially to the sparseness $k$, which is unavoidable with the present set of assumptions but undesirable in practice.
- The construction of the hard instance depends on exponentially large state space. This is not uncommon in the theoretical analysis, but worth discussing if similar results hold with smaller state space.

Some mathematical definitions are confusing:
- Def of $\Delta(\cdot)$ lacking?
- Why parameters restricted on sphere? (I suspect the authors are confusing sphere and ball)
- How s_0 is sampled? Is it fixed?

**Questions:**

Does the proposed method exploit the sparseness except through the cardinality of the epsilon net?

---

> ### Author Response · Authors · 2024-11-23
>
> Thank you for your careful review! We address your questions below:
>
> **Practicability:** In this paper, we focus on the setting where the sparsity $k$ is much smaller than the feature dimension $d$. In particular, when the sparsity $k$ is a constant, our algorithm has a polynomial dependency on $H$ and $d$. This is because we are considering a much smaller epsilon-net that exploits the sparsity structure, instead of an epsilon-net for the whole feature space. For instance, when the optimal Q-function is a $d=100$ dimensional linear function, but can be approximated by a $k=2$-sparse linear function, the sample complexity and running time of our algorithm would be quadratic and could be practical for real-life applications.
>
> **Exponentially large state space:** When state space is small, we can directly use algorithms in tabular RL setting and get arbitrarily small suboptimality with polynomial sample complexity, using, for example, the algorithm in Jin et al.(2018). Therefore, we cannot construct a hard instance with a small state space.
>
> **$\Delta(\cdot)$:** $\Delta(S)$ is the set of all distribution over $S$. We have added in this definition in the preliminary section in the revised version. Thank you for pointing it out.
>
> **Sphere vs. Ball:** In Assumption 1, we stated that each optimal parameter $\theta_h^*$ is in the d-dimensional unit sphere $\mathbb{S}^{d-1}$. Our results can be extended to assuming $\theta_h^*$’s are in the unit ball, i.e. $\|\theta_h^*\|\leq 1$ by creating a grid for the radius.
>
> **$s_0$:** Yes. We assumed the initial state $s_0$ is deterministic at line 229.
>
> **Question:** In the upper bound result, we agree that we mainly exploit the sparseness through the cardinality of the set of non-zero indices and through the cardinality of the epsilon net. Therefore, the results supersede a setting where the misspecified Q-function of each layer is in a finite-sized function class as a special case. Assuming there are $F$ functions at each layer, we can construct the feature vector as $F$-dimension, where the i-th index is the value of the i-th function. Then, this setting is captured by a $1$-sparse parameter $\theta^*$.
>
> References:
> Chi Jin, Zeyuan Allen-Zhu, Sebastien Bubeck, Michael I. Jordan. *Is Q-learning Provably Efficient?* Advances in Neural Information Processing Systems 31 (NeurIPS 2018)

---

> > ### Comment · Reviewer_AMgy · 2024-11-24
> >
> > Thanks for the clarification. I will keep my score.

---

### Official Review · Reviewer_5fDB · 2024-11-02

**Soundness:** 4
**Presentation:** 4
**Contribution:** 2
**Rating:** 6
**Confidence:** 4

**Summary:**

This paper considers the problem of online reinforcement learning with linear function approximation. It is well known if that in the presence of epsilon-misspecification (in the sense that Q* is eps-approximately linear), the best one can hope for is a an error guarantee that scales with \sqrt{d}*eps, where $d$ is the feature dimension, even in the bandit setting where $H=1$. The main result in this paper is to show that if we additionally assume that the optimal parameter is $k$ sparse, it is possible to improve upon this guarantee. In particular:

* 1) The author show that it is possible to achieve an error guarantee that scales as H*eps (with no dependence on dimension), albeit with sample complexity and runtime scaling as roughly (d/eps)^{k}.

* 2) The authors show that the error guarantee above cannot be improved further, in the sense that achieving H/T*eps for any T requires exp(T) sample complexity.

**Strengths:**

This paper makes a reasonable contribution to the theory of reinforcement learning. Guarantees in the linearly (approximately) realizable setting the authors consider are non-trivial and often somewhat surprising, so I think this paper makes an interesting contribution by filling in another piece of the landscape.

**Weaknesses:**

The main drawbacks preventing me from giving a higher score are as follows:

* On the upper bound side, the significance of the algorithmic techniques is somewhat narrow. Concretely, the authors consider a setting where we have a separate parameter $\theta*_h$ for each step $h\in[H]$. If we instead assume that there is a shared parameter $\theta*$ for all steps $h$, then I believe that simply enumerating over an epsilon-net for the parameter space and trying each policy one-by-one would be sufficient to achieve the approximation and sample complexity guarantees the authors get. So, the only reason why a non-trivial algorithm is required at all is because there is no parameter sharing across the steps $h$. Indeed, without parameter sharing, an epsilon-net would have size at least 2^H and lead to 2^H sample complexity, so the authors' contribution can be viewed as removing this exponential dependence on $H$ for this setting.

* The problem setting in the paper is fairly niche, and will probably only be interesting/compelling to research working on core RL theory. Broader impact is unclear.

**Questions:**

Overall, I do think the results are interesting and not necessarily obvious a-priori---they certainly piqued my curiosity. Some questions I was inspired to think about based on the results are as follows:

* Can we get similar guarantees beyond linear function approximation? My thought would be that if we assume that Q*_h is eps-approximately realizable by a class \mathcal{F}_h, then perhaps we can get H*eps error while paying for roughly max_h{Covering-Number(\cF_h, eps)} in the sample complexity using a generalization of the algorithm. If so, this would suggest that the linear structure is actually not that important.

* Have the authors thought about whether there exists an algorithm that achieves the H/T*eps error vs exp(T) sample complexity tradeoff in the lower bound?

* Is there any hope of achieving just d^{k} sample complexity instead of (d/eps)^k sample complexity?

---

> ### Author Response · Authors · 2024-11-23
>
> Thank you for your careful review and positive evaluation! Below, we respond to all your questions:
>
> **Weakness1:** If the parameters are shared across levels, then the problem setting can only apply to a very limited number of scenarios. Because of the sparsity assumption, such a setting only has $k$ non-zero parameters. On the other hand, our setting can be applied to a much larger range of problems since we have $Hk$ non-zero parameters.
>
> **Weakness2:** We believe core RL theory is an important research topic :)
>
> **Question1:** Yes. The function class setting can be thought of as a special case of our linear setting. When there are $F$ functions at each layer, we can construct the feature vector as $F$-dimension, where the i-th index is the value of the i-th function. Then, this setting is captured by a $1$-sparse parameter $\theta^*$.
>
> **Question2:** If the transition is deterministic, then we could modify our Algorithm 1 to achieve $H/C \cdot \epsilon$ vs. $\exp(C)$ trade-off. Specifically, we first separate the $H$ levels into blocks, each consisting of $C$ levels. Within each block, we would traverse through all the possible actions, with in total $a^C$ trajectories. Then, we eliminate each block based on the smallest Bellman error in these trajectories. This process ensures that we incur at most $\epsilon$ suboptimality within each block, leading to $O(H/C \epsilon)$ suboptimality. The total sample complexity scale by $a^C$.
>
> If the transition is stochastic, we might be able to use the importance sampling idea of Kearns et al. (1999) in our setting. Such a generalization could be nontrivial and we will leave it as a future work.
>
> However, restricting the sample complexity to be polynomial in this setting forces $C$ to be a log factor, resulting in only a log improvement in suboptimality. As this improvement is relatively minor and risks complicating the presentation, we chose not to include these results in the paper.
>
> **Question3:** Previous works (Weisz et al., 2022, Wang et al., 2021) show that even when the optimal Q-function is well-specified, any RL algorithm would have exponential dependency on $d$ or $H$. Note that this is equivalent to the case where the sparsity $k = d$ and the approximation error $\epsilon = 0$ in our setting. Therefore, in our misspecified setting which is strictly harder, exponential dependency on $k$ is unavoidable, unless we can accept an exponential dependency on $H$. However, we agree that a better dependency on $k$, such as $O(2^k)$ rather than $\epsilon^{-k}$, might be possible. We have added such improvements as future work.
>
> References:
>
> Michael Kearns, Yishay Mansour, Andrew Ng. *Approximate Planning in Large POMDPs via Reusable Trajectories.* Advances in Neural Information Processing Systems 12 (NIPS 1999)
>
> Gellert Weisz, Csaba Szepesvari, and Andras Gyorgy. TensorPlan and the Few Actions Lower Bound for Planning in MDPs under Linear Realizability of Optimal Value Functions. 2022.
>
> Yuanhao Wang, Ruosong Wang, and Sham M. Kakade. An Exponential Lower Bound for Linearly-Realizable MDPs with Constant Suboptimality Gap. 2021

---

> > ### Comment · Reviewer_5fDB · 2024-11-23
> >
> > Thank you for answering my questions! I will maintain my positive score.

---

### Official Review · Reviewer_5nER · 2024-11-04

**Soundness:** 3
**Presentation:** 3
**Contribution:** 3
**Rating:** 6
**Confidence:** 3

**Summary:**

The authors found a misspecified sparse linear bandit algorithm based on elimination. They first proved that the traditional approaches such as OLIVE work suboptimally in this environment, and also a naive extension from the previous bandit work [Dong & Yang, 2023] fails. After that, the authors propose a lower bound from the MDP without sample to RL with sample, using the INDQ problem as a mediator. This lower bound matches with the upper bound of their algorithm, showing that their algorithm has a near-optimal rate.

**Strengths:**

1) Clarity & Soundness: Their writing was clear on these points.
- They clearly stated their novelty and significant difference from the misspecified bandit algorithm proposed by Dong & Yang.
- They tried to demonstrate how they set the lower bound instance by high-level, but an explicit explanation throughout Section 3, with a nice figure. (and also the main difference from Du et al. 2020)
- The algorithm itself is also quite simple - it follows a traditional method of elimination algorithms, and thanks to the detailed explanation, I could easily get a sketch about how the proof goes on. The technique is also using some simple Hoeffding's inequality.

2) Significance & Novelty: Even though their results are clearly written and easy to understand, many of their results are quite fresh and counter-intuitive. One cannot extend bandit algorithms in this case was quite interesting indeed.

**Weaknesses:**

1) Computationally intractable. If I understood correctly, they use an $\epsilon$-net on the candidates for $\theta_h$ - $P_h^0$ starts from all possible $k$-sparse vectors. After that, for each iteration, this algorithm is searching for argmax over all candidates. Also, eventually they proposed an algorithm that provides $\epsilon$-optimal.

2) Even though they tried to show their process transparently, there are some parts that I cannot understand. Please check the 'Questions' below.

**Questions:**

1) I don't understand the word 'MDP without sample.' Without any trajectory, how could a learning agent learn something? You mean, 'without learning' the error is bounded by $\Omega(H\epsilon)$?

2) Also, it is interesting that this algorithm eliminates the 'most promising candidate' first. In bandits, the elimination eliminates all the suboptimal candidates that show less upper bound than the largest lower bound (eliminate an arm i which is max_j LCB_t(j) > UCB_t (i)). Why does this difference happen?

3) So what is the optimal size of the parameters, $\epsilon_{net}$ and $\epsilon_{stat}$? I guess they also should be in the scale of $\epsilon$.

3-1) Also just to clarify, what is $\epsilon$-optimal in here, does this mean $V^* - V(\pi) \leq \epsilon$? If so, according to your final result (Lemma 4.4), for the error $\epsilon'$, one should set $\epsilon/H$. This means now $m$ is proportional to $H^2$ too and the overall sample complexity also should include $H^2$, I believe.

3-2) In that sense, the line 438 is very confusing. Why it is $H^2 \cdot m$, including the $\epsilon_{stat}^2$ in the denominator? You repeat $m$ samplings for $H$ iterations. Shouldn't it be $H\cdot m$?

---

> ### Author Response · Authors · 2024-11-23
>
> Thank you for your careful review and the positive score! Below, we respond to all your questions:
>
> **Computationally intractable:**  In this paper, we focus on the setting where the sparsity $k$ is much smaller than the feature dimension $d$. In particular, when the sparsity $k$ is a constant, our algorithm has polynomial dependency on $H$ and $d$. This is because we are considering a much smaller epsilon-net that exploits the sparsity structure, instead of an epsilon-net for the whole feature space. For instance, when the optimal Q-funcion is a $d=100$ dimensional linear function, but can be approximated by a $k=2$-sparse linear function, the sample complexity and running time of our algorithm would be quadratic.
>
> **“MDP without sample”:** This setting serves as a warmup for the more complicated problem instance construction in the next section. Moreover, since our hard instance is 1-dimensional, the feature $\phi$ is equal to the optimal Q-value up to an error of $\epsilon$, so it is a very simple setting where an approximate version of the optimal Q-function is given to the algorithm. Our lower bound indicates that, even in this very simple setting, a suboptimality of $O(H\epsilon)$ is the best one can hope for. Therefore, the problem we are studying here is:
> - if we are given a perturbed optimal $Q$-function, without further learning, how does the perturbation translate to the suboptimality of the resulting policy in the minimax sense.
>
> **Eliminating most promising candidate:** In UCB of the Bandit setting, arms are eliminated based on the confidence bound of *each arm's reward*. In our algorithm, we maintain a set of all the remaining *parameters*, select the parameters with the largest value at each iteration, and eliminate them if the Bellman Error (Def. 4.1, measuring consistency between parameters of two consecutive horizons) is large. The two algorithms follow completely different frameworks, one maintains a set of remaining arms, and the otehr maintains a set of possible parameters.
>
> **Optimal sizes of $\epsilon_{net}$ and $\epsilon_{stat}$:** The optimal value of $\epsilon_{net}$ and $\epsilon_{stat}$ depends on users' priorities. A larger value of $\epsilon_{net}$ and $\epsilon_{stat}$ leads to a larger number of iteration (Lemma 4.1) and a larger data requirement (Line 448). However, it leads to a smaller suboptimality with high probability (Line 459-461). Therefore, there is no universal optimal value of $\epsilon_{net}$ and $\epsilon_{stat}$. As a rule of thumb, if one wants the final suboptimality to scales with $\epsilon$, then $\epsilon_{net}$ and $\epsilon_{stat}$ should be in the scale of $\epsilon$.
>
> **$\epsilon$-optimal:** We define $\epsilon$ as the misspecification error (Assumption 1.1). If we see $\epsilon$ as the suboptimality (i.e. $V(\pi^*) - V(\pi) \leq \epsilon$) and use a separate notation for the misspecification error, then the sample complexity would indeed need to scale by $H^2$.
>
> However, as indicated by our lower bounds (Theorem 3.1 and 3.2), we are unable to achieve less than $O(H\epsilon)$ suboptimality (here $\epsilon$ is the misspecification error), unless we take a sample size that is exponential in $H$. We have added a footnote to clarify this.
>
> **$H^2m$:** We apologize for the typo. In each iteration of the algorithm, we draw a new sample of $m$ trajectories for the estimation at each level, so we need $Hm$ samples per iteration.

---

### Meta-Review · Area_Chair_MM5G · 2024-12-23

**Metareview:**

This paper present a elimination-based algorithm and polynomial sample complexity bound for reinforcement learning with sparse linear function approximation. The sample complexity is polynomial in the feature dimension d and planning horizon H when the sparsity level is constant. A lower bound matching to logarithmic factors is also provided to complement the result. The problem is important and technical result is solid. All reviewers are supportive of this work, we thus recommend acceptance.

**Additional Comments On Reviewer Discussion:**

All reviewers support the acceptance of this paper.

---

### Decision · Program_Chairs · 2025-01-22

Accept (Poster)